# Suppressive IL-17A$^+$Foxp3$^+$ and ex-Th17 IL-17A$^{neg}$Foxp3$^+$ T$_{reg}$ cells are a source of tumour-associated T$_{reg}$ cells

Stephanie Downs-Canner[1,*], Sara Berkey[1,*], Greg M. Delgoffe[2,3,4], Robert P. Edwards[2,4,5], Tyler Curiel[6], Kunle Odunsi[7], David L. Bartlett[1] & Nataša Obermajer[1]

Th17 and regulatory T (T$_{reg}$) cells are integral in maintaining immune homeostasis and Th17–T$_{reg}$ imbalance is associated with inflammatory immunosuppression in cancer. Here we show that Th17 cells are a source of tumour-induced Foxp3$^+$ cells. In addition to natural (n)T$_{reg}$ and induced (i)T$_{reg}$ cells that develop from naive precursors, suppressive IL-17A$^+$Foxp3$^+$ and ex-Th17 Foxp3$^+$ cells are converted from IL-17A$^+$Foxp3$^{neg}$ cells in tumour-bearing mice. Metabolic phenotyping of Foxp3-expressing IL-17A$^+$, ex-Th17 and iT$_{reg}$ cells demonstrates the dissociation between the metabolic fitness and the suppressive function of Foxp3-expressing T$_{reg}$ cell subsets. Although all Foxp3-expressing subsets are immunosuppressive, glycolysis is a prominent metabolic pathway exerted only by IL-17A$^+$ Foxp3$^+$ cells. Transcriptome analysis and flow cytometry of IL-17A$^+$Foxp3$^+$ cells indicate that Folr4, GARP, Itgb8, Pglyrp1, Il1rl1, Itgae, TIGIT and ICOS are Th17-to-T$_{reg}$ cell transdifferentiation-associated markers. Tumour-associated Th17-to-T$_{reg}$ cell conversion identified here provides insights for targeting the dynamism of Th17–T$_{reg}$ cells in cancer immunotherapy.

[1] Department of Surgical Oncology, University of Pittsburgh, 5150 Centre Avenue, Pittsburgh, Pennsylvania 15213, USA. [2] University of Pittsburgh Cancer Institute, 5150 Centre Avenue, Pittsburgh, Pennsylvania 15213, USA. [3] Tumour Microenvironment Center, Hillman Cancer Center, University of Pittsburgh, 5115 Centre Avenue, Pittsburgh, Pennsylvania 15213, USA. [4] Magee-Womens Research Institute Ovarian Cancer Center of Excellence, 5150 Centre Avenue, Pittsburgh, Pennsylvania 15213, USA. [5] Peritoneal/Ovarian Cancer Specialty Care Center, Pittsburgh, Pennsylvania 15213, USA. [6] UT Health Science Center at San Antonio, 8403 Floyd Curl Drive, San Antonio, Texas 78229, USA. [7] Departments of Gynecologic Oncology and Immunology, Roswell Park Cancer Institute, Elm and Carlton Streets, Buffalo, New York 14263, USA. * These authors contributed equally to this work. Correspondence and requests for materials should be addressed to N.O. (email: nobermaj@its.jnj.com).

Regulatory T ($T_{reg}$) cells expressing the transcription factor forkhead box P3 (Foxp3), most of which are CD4$^+$ T cells that express CD25 (the interleukin-2 (IL-2) receptor α-chain), are indispensable for the maintenance of dominant self-tolerance and immune homeostasis, but also suppress antitumour immune responses and favour tumour progression. Tumour-induced expansion of $T_{reg}$ cells is a critical obstacle to successful cancer immunotherapy[1] and $T_{reg}$ cells are the subject of intense investigation as a primary target in the search for new therapeutic modalities. The manipulation of $T_{reg}$ cells is a crucial component of tumour immune surveillance and is based on numerous approaches, including depletion, reducing survival or suppressing the function of $T_{reg}$ cells with tyrosine kinase inhibitors, low-dose cyclophosphamide and paclitaxel, as well as checkpoint inhibitors and IL-2Rα-targeting agents[2]. Studies that target $T_{reg}$ cells in patients with cancer are limited, however, by the lack of an exclusive targetable surface molecule expressed on $T_{reg}$ cells.

There has been considerable debate in the field[3-6] regarding the concepts of Foxp3$^+$ $T_{reg}$ cell plasticity[7] and instability[8-10]. In plastic $T_{reg}$ cells the core $T_{reg}$ cell identity (Foxp3 expression and suppressive capacity) is maintained, but their malleable nature allows phenotypic and functional adaptation[7]. In contrast, $T_{reg}$ cell instability is marked by the loss of Foxp3 expression and suppressive capacity as well as acquisition of features reminiscent of effector T cells by ex-$T_{reg}$ cells in response to environmental cues[8-10]. The plasticity and instability of $T_{regs}$ cells has important therapeutic implications for the targeting of $T_{reg}$ cells. Although natural (n)$T_{reg}$ cells are usually stable and long-lived[3], $T_{reg}$ cells may demonstrate instability under pathogenic or inflammatory circumstances[4]. $T_{reg}$ cell instability has been detected in patients with colon cancer wherein Foxp3$^+$RORγt$^+$ IL-17-producing pathogenic cells[11] presumably arise from Foxp3$^+$ $T_{reg}$ cells that retain their suppressive, but lose their anti-inflammatory, function. That IL-17-producing T cells are absent in the thymus is evidence that IL-17$^+$Foxp3$^+$ cells are generated in the periphery, confirming that instability is marked by a response to environmental cues[12].

$T_{reg}$ cell development and survival are dependent on a number of factors and signals, including IL-2, transforming growth factor-β (TGF-β) and co-stimulatory molecules (such as CD28). Cancer presents a favourable environment for inducing and maintaining $T_{reg}$ cell identity, by stimulating the $T_{reg}$ cell signature in de novo generated induced (i)$T_{reg}$ cells (derived from converted CD25$^-$ cells) and recruiting n$T_{reg}$ cells to the tumour site, both contributing to the pool of tumour-associated $T_{reg}$ cells. During resolution of inflammation, T helper type 17 (Th17) cells were shown to transdifferentiate into another regulatory T-cell subset, IL10$^+$ T regulatory type 1 (Tr1) cells[13]. An additional source of $T_{reg}$ cells includes Th17 cell transdifferentiation into ex-Th17 IL-17A$^{neg}$Foxp3$^+$ cells, described in an allogeneic heart transplantation model[14].

Here we characterize tumour-associated Th17-to-$T_{reg}$ cell transdifferentiation as an alternative source for tumour-associated $T_{reg}$ cells. Our data demonstrate that tumour-induced Th17 cells progressively transdifferentiate into IL-17A$^+$Foxp3$^+$ and ex-Th17 IL-17A$^{neg}$Foxp3$^+$ T cells during tumour development. We identify several Th17–$T_{reg}$ transdifferentiation-associated transmembrane molecules on IL-17A$^+$Foxp3$^+$ cells that may be feasible targets to manipulate $T_{reg}$ cell-associated tumour immune surveillance, and complement programmed cell death protein 1 (PD1)-mediated control of T-cell activation. Furthermore, the differences in the bioenergetic profiles of exTh17 IL-17A$^{neg}$Foxp3$^+$ and IL-17A$^+$Foxp3$^+$ or IL-17A$^+$ Foxp3$^{neg}$ cells offer an alternative method to steer plastic Th17 cells away from the $T_{reg}$ phenotype via metabolic reprogramming[15]. Finally, an increase in plastic Foxp3$^+$ Th17 cells infiltrating the tumour micorenvironment of ovarian cancer patients and the tumour-associated induction of Foxp3 expression in human IL-17A-producing ovarian cancer tumour-associated lymphocytes (TALs) validates the concept that inhibiting Th17-to-$T_{reg}$ cell conversion may serve as a valuable targeting strategy in tumour immunotherapy.

## Results

**IL-17A$^{neg}$Foxp3$^+$ ex-Th17 $T_{reg}$ cell emergence in cancer.** Th17 cells have considerable plasticity and readily shut off IL-17 production and shift into Th1-like cells in autoimmune and other chronic inflammatory disorders[16-19]. We investigated whether the cancer microenvironment affects Th17 cell stability in vivo. We crossed Il-17a$^{Cre}$ mice with Rosa26$^{eYFP}$ reporter mice to generate an Il-17a$^{Cre}$Rosa26$^{eYFP}$ mouse strain, in which the fluorescent reporter permanently labels Il-17a$^{Cre}$ cells. This allows the identification of cells that have switched on IL-17 expression (marked by enhanced yellow fluorescence protein (eYFP) expression)[20]. We show that eYFP expression (percentage of eYFP$^+$ cells of CD3$^+$CD4$^+$ cells, Fig. 1a and Supplementary Fig. 1a) gradually increases in tumour-bearing (ID8 ovarian cancer and MC38 colorectal cancer) mice over time. Cancer-associated Th17 cell induction is observed in the ovarian cancer ascites-infiltrating cells (TALs, $n = 4$ per group) and colorectal tumour-infiltrating cells (TILs, $n = 10$ per group), as well as in the spleens of tumour-bearing mice (Fig. 1a). These data reveal that cancer progressively induces Th17 cells in vivo. However, in contrast to the increase in the percentage of eYFP$^+$ cells, the production of IL-17A—after the initial increase—declines at later time points (Supplementary Fig. 1b). Fate determination of the cells from tumour-bearing mice that had produced IL-17A (ex-Th17 cells) shows that a considerable proportion of eYFP$^+$ cells begin expressing Foxp3 and the percentages of ex-Th17 Foxp3$^+$ CD4$^+$ T cells gradually increases (Fig. 1b and Supplementary Fig. 1c), while the percentage of eYFP$^+$Foxp3$^{neg}$ cells (that is, 'true' Th17 cells) declines at the later time points (Supplementary Fig. 1d). This concomitant decrease in the 'true' Th17 cells and emergence of ex-Th17 IL-17A$^{neg}$Foxp3$^+$ cells in the course of tumour progression provides evidence for the transdifferentiation of tumour-induced Th17 cells into IL-17A$^{neg}$Foxp3$^+$ in the tumour microenvironment.

**Changes in Foxp3$^+$ $T_{reg}$ cells in the tumours of RORγt$^{-/-}$ mice.** The orphan nuclear receptor RORγt expressed by Th17 cells is the key transcription factor that orchestrates the differentiation of Th17 cell lineage[21]. Th17 cells are absent in mice deficient in RORγt. The percentage of tumour-associated Foxp3$^+$ cells (Fig. 2a) and Foxp3 expression (Supplementary Fig. 2a) is significantly reduced in RORγ$^{-/-}$ ID8 tumour-bearing mice compared with controls ($n = 5$ per group), suggesting that Th17 cell transdifferentiation serves as an important pathway of $T_{reg}$ cell emergence in the tumour microvenvironment. However, the percentage of Foxp3$^+$ $T_{reg}$ cells in spleens of these mice is not changed. These data indicate that while $T_{reg}$ cell development in homeostasis is independent of Th17 cell induction, the RORγt pathway is involved in tumour induction of Foxp3$^+$ T cells. A reduced percentage of $T_{reg}$ cells in the tumour micro-environment is accompanied by a lower percentage of myeloid-derived immunosuppressive cells (MDSCs) (Supplementary Fig. 2b), demonstrating a significant change in the tumour environment accompanying the perturbed Th17–$T_{reg}$ cell developmental axis, and in line with the recent observations that specific ablation of RORγt in myeloid compartment impairs the generation of suppressive MDSCs[22].

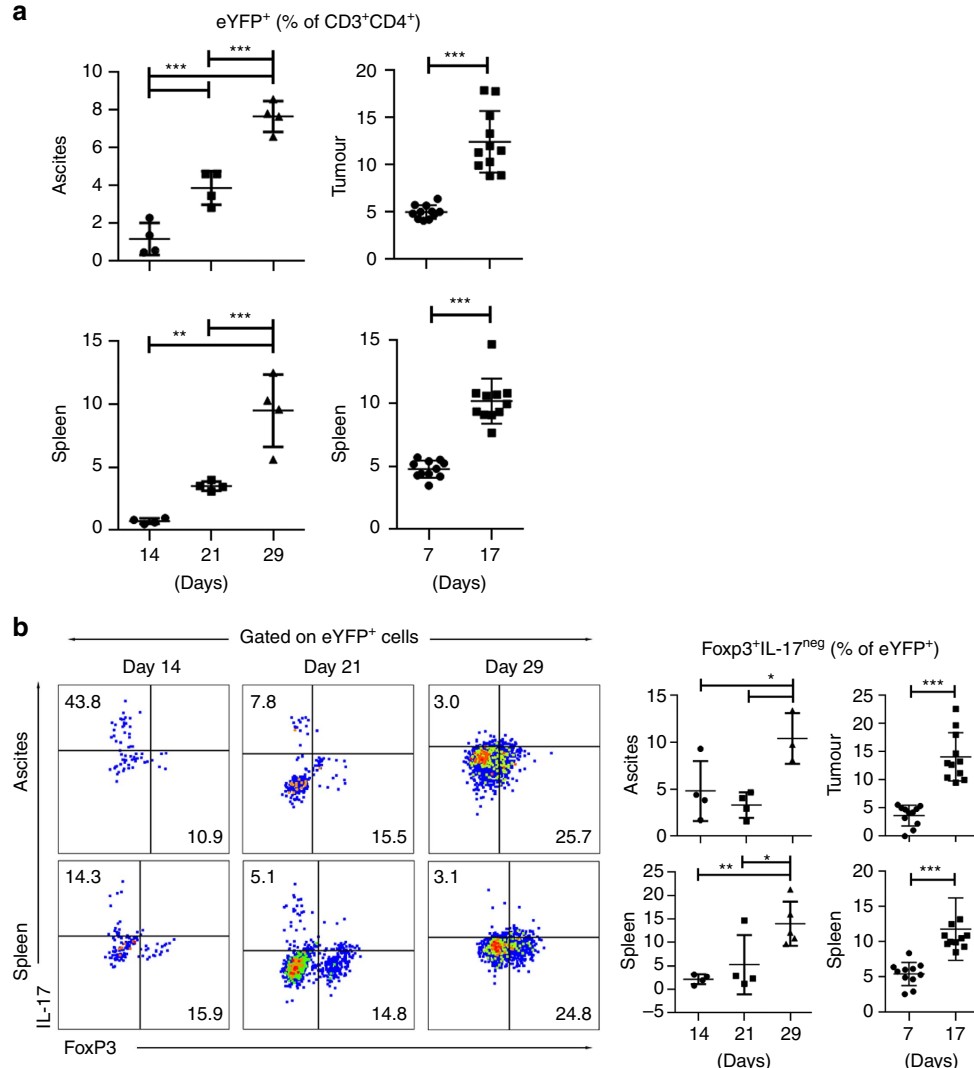

**Figure 1 | Th17 cells transdifferentiate into *Foxp3⁺exTh17* cells in tumour-bearing mice.** ID8A ovarian cancer and MC38 colorectal cancer cells were injected intraperitoneally in IL-17a$^{Cre}$R26R$^{ReYFP}$ fate reporter mice. TALs (ovarian cancer, $n = 4$ mice per group), TILs (colorectal cancer, $n = 10$ mice per group) and splenocytes were recovered at different time points of tumour progression and CD4$^+$ T cells assessed for eYFP expression (**a**) as well as eYFP$^+$ cells analysed for IL-17 production and Foxp3 expression (**b**) by flow cytometry. The fluorescence-activated cell sorting (FACS) gating strategy of live eYFP$^+$ CD4$^+$ T cells is presented in Supplementary Fig. 8a. We observed accumulation of eYFP$^+$ cells (indicative of Th17 and/or exTh17 cells) over time (**a**) and the percentages of *Foxp3$^+$IL-17A$^-$* (that is, exTh17 T$_{reg}$) cells increased in these mice with tumour progression. All data are mean ± s.d. *$P < 0.05$, **$P < 0.01$ and ***$P < 0.001$ by two-tailed Student's *t*-test. Similar results were obtained in an additional independent experiment.

In addition to reducing the Foxp3$^+$ T cells in the tumour microenvironment, our data reveal that the Th17-supporting transcription factor RORγt profoundly alters the phenotype of tumour-associated Foxp3$^+$ T cells (Fig. 2b,c). Helios, an Ikaros zinc-finger transcription factor, is involved in T$_{reg}$ cell development and stability and defines a highly suppressive T$_{reg}$ subset[23]. Foxp3$^+$Helios$^+$ T$_{reg}$ cells are significantly expanded in tumour microenvironments[24,25]. Compared with the Foxp3$^+$ CD4$^+$ T cells in ovarian cancer ascites of wild-type mice, the percentage of Helios$^+$ subset of Foxp3$^+$ cells infiltrating cancer ascites of RORγt$^{-/-}$ mice is significantly reduced (Fig. 2b). However, Foxp3$^+$ cells in the spleens of RORγt$^{-/-}$ mice are all Helios$^+$ and no differences with wild-type mice can be observed. The data reveal the importance of RORγt in the tumour-associated induction of a distinct T$_{reg}$ phenotype, and suggest a difference in the development of Foxp3$^+$ T$_{reg}$ cells in the setting of tumour compared with other microenvironments. Indeed, RORγt has been shown to have a central role in determining the balance

between protective and pathogenic T$_{reg}$ cells in colon cancer[11], where RORγt-expressing T$_{reg}$ cells, unable to suppress inflammation, expand in a cancer stage-dependent manner, and are directly associated with the amount of inflammation and disease progression[11]. Ablating RORγt specifically in T$_{reg}$ cells was shown to stabilize their anti-inflammatory properties[11,26].

Apart from Helios, the percentage of PD1$^+$ Foxp3$^+$ CD4$^+$ cells in ovarian cancer of RORγt$^{-/-}$ mice (42.4 ± 2.97% (control) and 24.55 ± 4.89 (RORγt$^{-/-}$), $n = 5$ per group) is significantly lower (Fig. 2c). The reduced expression of PD1 is specific for Foxp3$^+$ CD4$^+$ cells, but not Foxp3$^{neg}$ CD4$^+$ cells (5.34 ± 1.55% (control) and 5.47 ± 1.88 (RORγt$^{-/-}$)) or CD4$^{neg}$ CD3$^+$ cells (21.48 ± 5.44% (control) and 16.94 ± 6.43 (RORγt$^{-/-}$)). The lack of PD1$^+$ T$_{reg}$ cells is associated with a loss of therapeutic benefit from PD1 blockade, with treated wild-type mice outliving treated RORγt$^{-/-}$ mice (Fig. 2d), while the survival of untreated RORγt$^{-/-}$ mice is not different compared with untreated wild type mice (Fig. 2d). This striking observation

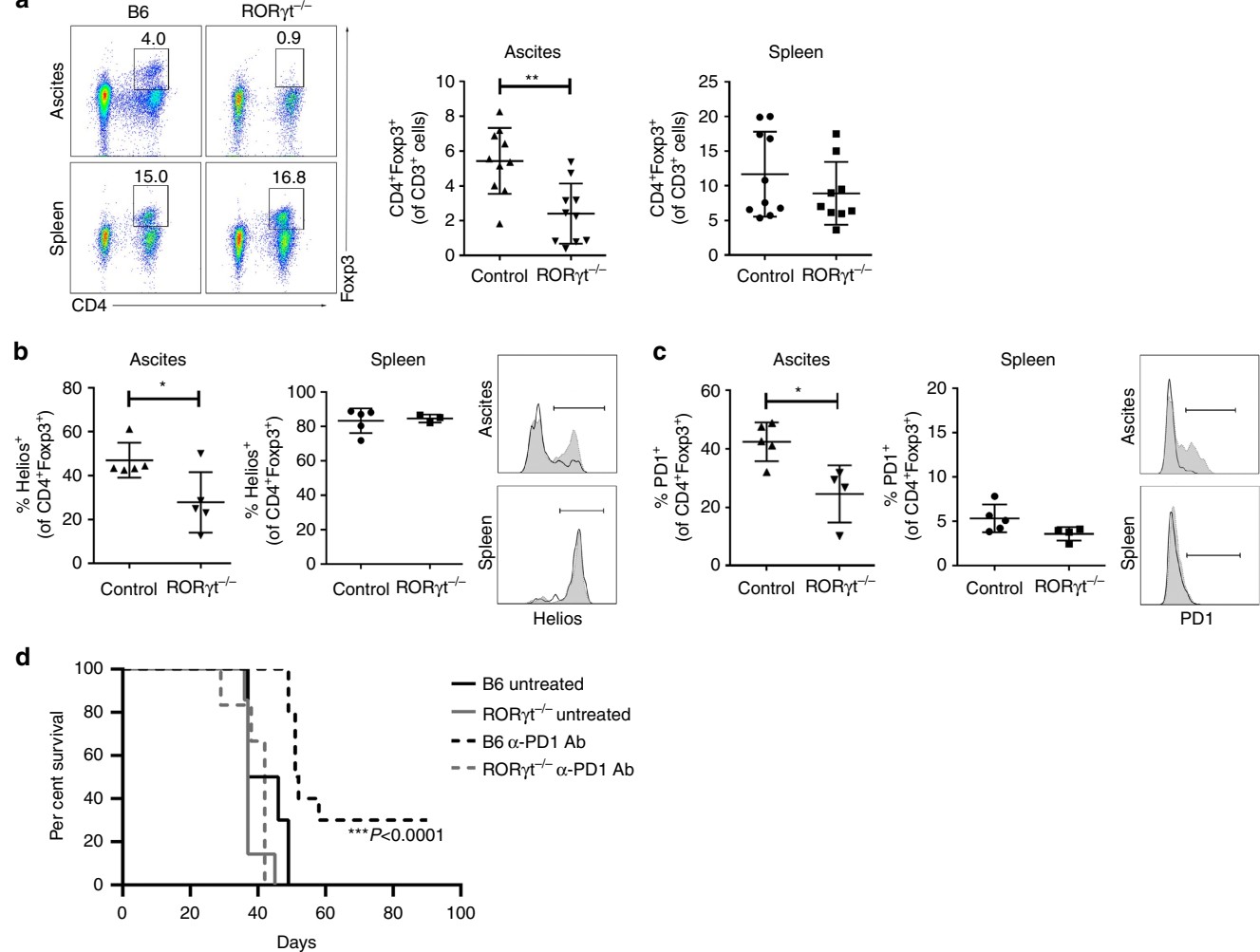

**Figure 2 | Changes in tumour-associated T$_{reg}$ cells in Rorγt$^{-/-}$ mice.** Rorγt$^{-/-}$ and B6 mice were injected i.p. with ID8A cells. (**a**) The percentage of Foxp3$^+$CD4$^+$ cells of CD3$^+$ cells in the tumour ascites and spleens were determined on day 35 ± 2. Representative staining (left) and statistical analysis (right) of pooled data from two independent experiments with $n = 5$ mice, respectively. The fluorescence-activated cell sorting (FACS) gating strategy of live T cells is presented in Supplementary Fig. 8b. (**b,c**) Percentages of Helios$^+$ (**b**, left) and PD1$^+$ (**c**, right) and representative staining of Foxp3$^+$CD4$^+$ cells in tumour ascites and spleens from $n = 5$ mice. Black line indicates the staining of CD4$^+$Foxp3$^+$ cells from B6 mice and grey filled line CD4$^+$Foxp3$^+$ cells from Rorγt$^{-/-}$ mice. (**d**) Survival time was monitored in untreated and α-PD1 antibody (Ab)-treated Rorγt$^{-/-}$ ($n = 5$) and B6 mice ($n = 10$). Mice were treated with either PBS (200 μl i.p.) or RMP1-14 (α-PD1 Ab, 5 mg kg$^{-1}$, BioXcell, 200 μl i.p.) on days 3, 6, 9 and 12. Spleens and ascites from untreated B6 and Rorγ$^{-/-}$ mice ($n = 5$ per group) were collected at day 35 ± 2 and cells purified. Survival curves were compared using log-rank (Mantel–Cox) test. All data (**a**–**c**) are mean ± s.d. *$P < 0.05$, **$P < 0.01$ and ***$P < 0.001$ by two-tailed Student's $t$-test.

prompted us to investigate the role of Th17 and T$_{reg}$ cell subsets in cancer progression.

**Ex-Th17 Foxp3$^+$ and IL17A$^+$Foxp3$^+$ cells are suppressive.** The role of fundamentally immunosuppressive T$_{reg}$ cells and proinflammatory Th17 cells is context dependent in the sterile (para-)inflammation (as defined by Medzhitov *et al.*[27]) of the tumour microenvironment. Not surprisingly, studies evaluating the clinical relevance of T$_{reg}$ and Th17 cells in cancer immunology have had contradictory results (reviewed in refs 28–30). It was reported that the ability of T$_{reg}$ cells to control cancer-associated Th17 cell-mediated inflammation is lost in the course of disease and T$_{reg}$ cells shift from a protective IL-10-producing anti-inflammatory to an IL17-producing cancer-promoting proinflammatory phenotype, with preserved capacity to suppress protective antitumour immune responses[31–33]. To examine the role of Th17 and T$_{reg}$ cell plasticity in tumour

progression, we induced Th17/T$_{reg}$ cell differentiation *in vitro* by culturing CD4$^+$ T cells from *Foxp3$^{GFP}$* mice under Th17–T$_{reg}$-driving conditions (Fig. 3a). We isolated IL-17A-producing Foxp3$^{neg}$, IL-17A-producing Foxp3$^+$ and IL-17A$^{neg}$ Foxp3-expressing CD4$^+$ T cells (Fig. 3b) and injected the Th17 and T$_{reg}$ subsets in Cd45.1 tumour-bearing mice on days 5, 12 and 19 ($n = 4$ per group). While IL-17A$^{neg}$Foxp3$^{neg}$ and IL-17A$^{neg}$ Foxp3$^+$ T$_{reg}$ cells do not affect tumour progression when compared with control tumour-bearing mice (Fig. 3c), both subsets of IL-17A-producing Foxp3$^+$ and Foxp3$^{neg}$ cells enhance tumour progression compared with IL-17A$^{neg}$Foxp3$^+$ T$_{reg}$ cells (Fig. 3c). Despite the apparent differences in tumour progression (Supplementary Fig. 3a) the survival of the mice is not significantly different between the groups and all mice need to be killed between days 40 and 45 (Supplementary Fig. 3b). We analysed the tumour ascites and spleens of these mice focusing on the adoptively transfered subsets, but not their effects on inherent innate and adoptive immune cells (Fig. 3d and Supplementary

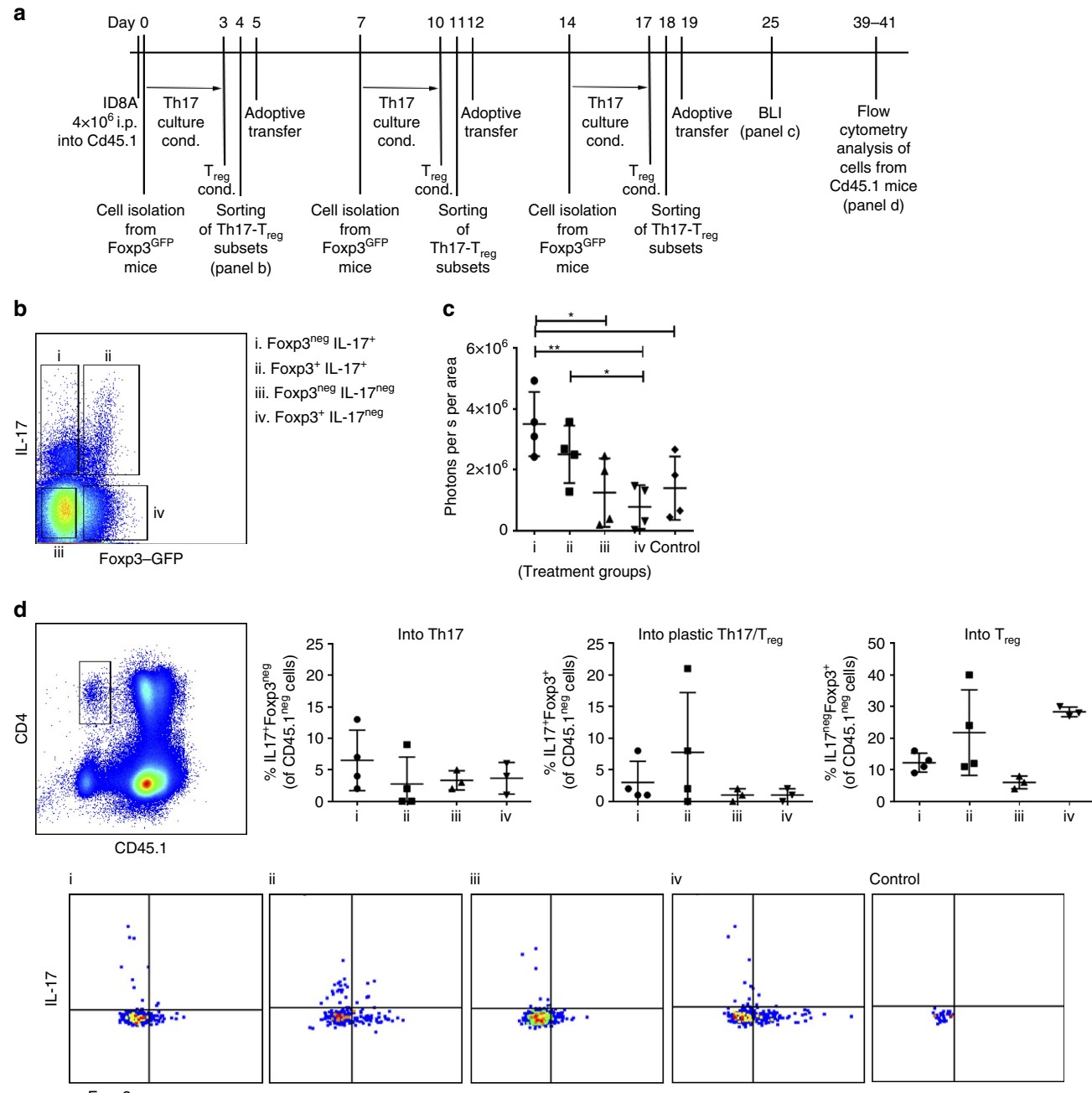

**Figure 3 | Transdifferentiation of adoptively transfered Foxp3$^{neg/+}$ IL-17A$^+$ T cells. (a)** Timeline of the experiment: CD4$^+$ T cells from *Foxp3$^{GFP}$* reporter mice were cultured under Th17 (IL-6, IL-23, TGF-β, 3 days)/T$_{reg}$ (TGF-β, +1 day)-driving conditions. Foxp3$^{neg}$IL-17A$^+$ (i), Foxp3$^+$IL-17A$^+$ (ii), Foxp3$^{neg}$IL-17A$^{neg}$ (iii) and Foxp3$^+$IL-17A$^{neg}$ (iv) CD4$^+$ T cells were sorted and injected i.p. into ID8A-tumour bearing mice on days 5, 12 and 19. **(b)** Following staining with IL-17A-detection kit, live/CD4$^+$/Th17-T$_{reg}$ subsets (i-iv) were flow sorted using the presented gating strategy (note that the 3 h activation with PMA and ionomycin before cell sorting results in a reduced mean fluorescence intensity of Foxp3-GFP; see Supplementary Fig. 9). The fluorescence-activated cell sorting (FACS) gating strategy for live CD4$^+$ cells is presented in Supplementary Fig. 8c. **(c)** Th17-T$_{reg}$ subsets (i-iv) were injected i.p. into ID8A-luc tumour-bearing Cd45.1 mice on days 5, 12 and 19 and tumour growth monitored by bioluminescence (day 25, $n = 4$ mice per group) compared with control tumour-bearing mice. **(d)** CD4$^+$CD45.1$^{neg}$ cells from ID8A-luc tumour-bearing Cd45.1 mice, receving adoptively transfered Th17-T$_{reg}$ subsets (groups i-iv, $n = 4$ mice per group), were analysed on day 40±1 for IL17A production and Foxp3 expression. Statistical analysis of the percentages of Foxp3$^{neg}$IL-17A$^+$, Foxp3$^+$IL-17A$^+$ and Foxp3$^+$IL-17A$^{neg}$ of CD4$^+$CD45.1$^{neg}$ cells in each group of mice (top) and representative staining (bottom). The FACS gating strategy of live cells is presented in Supplementary Fig. 8d. All data **(c,d)** are mean ± s.d. *$P < 0.05$ and **$P < 0.01$ by two-tailed Student's t-test.

Fig. 3c). Analysis of CD45.1$^{neg}$ cells revealed substantial IL-17A$^+$Foxp3$^{neg}$ and IL-17A$^+$Foxp3$^+$ cell plasticity. Specifically, 12.25% (9–16%) of IL-17A$^+$Foxp3$^{neg}$ and 21.75% (11–40%) of IL-17A$^+$Foxp3$^+$ cells transdifferentiate into ex-Th17 Foxp3$^+$ T cells (Fig. 3d). IL-17A$^{neg}$Foxp3$^+$ T cells

however do not demonstrate a strong plastic potential and only 3.67 and 1.0% of the cells convert into IL-17A$^+$Foxp3$^{neg}$ and IL-17A$^+$Foxp3$^+$ cells, respectively, whereas 28.3% (27–30%) of the cells remain IL-17A$^{neg}$ Foxp3-expressing cells (Fig. 3d). This observed limited plasticity of Foxp3$^+$ T$_{reg}$ cells is in line with

recent reports documenting the selective differentiation of IL-17A$^+$Foxp3$^+$ T cells only from a population of CCR6$^+$ memory Foxp3$^-$ and CCR6$^+$ Foxp3$^+$ T cells[34,35]. Our data show that both IL-17A$^+$Foxp3$^{neg}$ and IL-17A$^+$Foxp3$^+$ cells convert into IL-17A$^{neg}$Foxp3$^+$ T cells. Although the number of the analysed cells are limited, the significance of the data is primarily the identification rather than quantification of ex-Th17 IL-17A$^{neg}$Foxp3-expressing cells.

Our data demonstrate that the role of a specific cell subset in a tumour microenvironement depends not only upon its functional capacity in any given moment, but also its plasticity potential. Therefore, the role of a plastic cell subset cannot be simply inferred from its phenotype at a specific time point, but instead, can be understood to exist on a continuum which is dependent upon the tumour microenvironment. This finding helps reconcile the contradictory information about the relevance of Th17 subset in cancer immune surveillance.

**TGF-β and PGE$_2$ foster Th17-to-T$_{reg}$ cell transdifferentiation**. TGF-β is a common denominator of mouse T$_{reg}$ and Th17 cell biology. In the presence of TGF-β, CD4$^+$ T cells are induced to express Foxp3 and are converted towards an iT$_{reg}$ cell fate[36]. Still however, TGF-β is also critical for mouse T-cell commitment to Th17 cell development[37,38]. With the goal of investigating the relationship of tumour-induced Th17-to-T$_{reg}$ cell conversion to *de novo* iT$_{reg}$ cell differentiation, we cultured splenocytes from *Il17a*$^{Cre}$*Rosa26*$^{eYFP}$ (Fig. 4a–d) and *Il17a*$^{Cre}$*Rosa26*$^{eYFP}$x*Foxp3*$^{tm1Flv}$ (Fig. 4e) mice under T$_{reg}$- and Th17-driving conditions, and in tumour cell-conditioned medium. CD4$^+$ eYFP$^+$ cells are present as early as day 3 (Fig. 4a,b) and their percentage increases further at later time points (days 5–7) under Th17-driving conditions (TGF-β, IL-6 and IL-23) (Fig. 4b). While the ex-Th17 Foxp3$^+$ subset is present at day 3 in both conditions, a distinct shift in the composition of eYFP$^+$ cells occurs by day 5 in cells cultured in T$_{reg}$ compared with Th17-driving culture conditions (Fig. 4c). TGF-β (T$_{reg}$-driving condition) promotes an increase in ex-Th17 Foxp3$^+$ cells and the presence of IL-6 promotes the expansion of true Th17 cells (IL-17$^+$Foxp3$^{neg}$). Similar data were obtained using the splenocytes from *Il17a*$^{Cre}$*Rosa26*$^{eYFP}$x*Foxp3*$^{tm1Flv}$ mice (Fig. 4e) cultured under Th17–T$_{reg}$-driving (Fig. 4e, top) and T$_{reg}$-driving (Fig. 4e, bottom) conditions, respectively. The YFP$^+$ and YFP$^{neg}$ Th17-T$_{reg}$ subsets from these mice were sorted for the assessment of their metabolic fitness and immunosuppressive functions (see Fig. 5). The analysis of the expression of RORγt and Helios transcription factors in YFP$^+$ and YFP$^{neg}$ Th17–T$_{reg}$ subsets generated under Th17–T$_{reg}$-driving conditions reveals that YFP$^{neg}$ cells do not express RORγt, whereas Helios is induced in YFP$^{neg}$Foxp3$^+$ subset. In contrast, YFP$^+$ Th17–T$_{reg}$ subsets do express both RORγt and Helios, but to a different extent (Supplementary Fig. 4a). The analysis of the RORγt and Helios expression over time shows that in YFP$^+$CD4$^+$ cells cultured under Th17-driving conditions the expression of RORγt is relatively uniform, while the expression of Helios declines over time (Supplementary Fig. 4b). The expression of PD1 is significantly induced in Foxp3$^{+/-}$ IL-17A-producing as well as Foxp3$^+$IL-17A$^{neg}$ subsets compared with Foxp3$^{neg}$IL-17A$^{neg}$ subset (Supplementary Fig. 4c). In line with the above data in RORγt$^{-/-}$ mice, we here further delineate a common initial developmental pathway which, depending upon context, diverges into apparent reciprocal developmental pathways for the generation of Th17 and iT$_{reg}$ cells[39].

The deviation towards ex-Th17 T$_{reg}$ cells (IL17A$^{neg}$Foxp3$^+$ of eYFP$^+$) is promoted under T$_{reg}$-driving conditions, that is, TGF-β and when the CD4$^+$ T cells are cultured in tumour cell conditioned medium (Fig. 4d). TGF-β is not the only factor produced by tumour cells promoting ex-Th17 T$_{reg}$ cells. PGE$_2$, another factor inducing Foxp3 expression and T$_{reg}$ cell function[40], proves to play a role in Th17-to-T$_{reg}$ cell conversion (Fig. 4d). Blocking TGF-β and PGE$_2$ production individually specifically inhibited Th17 into IL-17$^{neg}$Foxp3$^+$ cell conversion (Fig. 4d and Supplementary Fig. 4d,e). However, no significant reduction in the percentage of IL-17$^{neg}$Foxp3$^+$ cells when both TGF-β and PGE$_2$ production were blocked demonstrates the complexity of the interplay between the two tumour-associated factors in promoting Th17 cell transdifferentiation. Furthermore, while the addition of TGF-β to IL-6 and IL-23 promotes IL-17A production by Th17 cells, the presence of PGE$_2$ overrides this effect and promotes the conversion of Th17 cells into IL-17A$^{neg}$Foxp3$^+$ cells (Supplementary Fig. 4d). Tumour-induced immunosuppressive factors in the tumour microenvironment play a major role in reprogramming Th cells.

**Suppressive IL17A$^+$Foxp3$^+$ cells are metabolically active**. The Th cell differentiation has been tightly linked to their intrinsic activity and function. To this end, Foxp3 in T$_{reg}$ cells is required and sufficient not only for T$_{reg}$ cell development, but also T$_{reg}$ cell immunosuppressive function[41–44]. To support their specific functional needs, functionally distinct T-cell subsets require distinct energetic and biosynthetic pathways. Indeed, compared with other CD4$^+$ T cells, T$_{reg}$ cells exhibit remarkable alterations in cellular metabolism, particularly in their nutrient substrate preference[45–47]. Specifically, Th1, Th2 and Th17 cells are highly glycolytic. T$_{reg}$, cells, in contrast, are thought to have high lipid oxidation rates *in vitro*[48]. We decided to determine whether the expression of T$_{reg}$-underlying transcription factor in the plastic IL-17A$^+$Foxp3$^+$ subset imprints a Foxp3-driven T$_{reg}$ cell gene expression program, imposing an immunosuppressive phenotype and function in the IL-17A$^+$Foxp3$^+$ cells. Furthermore, because the T$_{reg}$ cell-specific catabolic fatty acid-driven metabolic signature integrates their differentiation and function, we studied the metabolic profile of IL-17A$^+$Foxp3$^+$ cells.

Using the XF Extracellular Flux analyzer we determined the oxygen consumption rate (OCR) and extracellular acidification rate to investigate oxidative substrate flux and glycolysis in these cells. Both YFP$^+$IL-17A$^{neg}$Foxp3$^+$ (exTh17) and YFP$^{neg}$IL-17A$^{neg}$Foxp3$^+$ (iT$_{reg}$) cells show low glycolysis rate (Fig. 5a) and have low glycolytic capacity (Supplementary Fig. 5a,b) similar to control Foxp3$^{neg}$IL-17A$^{neg}$ cells. In contrast, YFP$^+$IL-17A$^+$Foxp3$^+$ cells have high rate of glycolysis that is comparable to the YFP$^+$IL-17A$^+$Foxp3$^{neg}$ cells (Fig. 5a and Supplementary Fig. 5a,b). This stress test reveals that the mitochondrial bioenergetic state of YFP$^+$IL-17A$^+$Foxp3$^+$ cells differs strikingly from YFP$^{+/neg}$IL-17A$^{neg}$Foxp3$^+$ cells and is similar to YFP$^+$IL-17A$^+$Foxp3$^{neg}$ cells. In addition to a higher basal OCR, the mitochondrial respiratory capacity (determined by FCCP (carbonyl cyanide-4-(trifluoromethoxy)phenylhydrazone)-stimulated OCR) is much higher in IL-17A$^+$Foxp3$^+$ cells, while the fraction of basal OCR contributing to ATP-coupled respiration (revealed by oligomycin-sensitive OCR) is similar to Foxp3$^{neg}$IL17A$^+$ cells (Supplementary Fig. 5c). Intriguingly, proton leak (the difference between oligomycin-resistant but rotenone-sensitive OCR) in Foxp3$^+$IL-17A$^+$ is markedly increased to 32% of the baseline OCR compared with 17% in Foxp3$^{neg}$IL-17A$^+$ cells. These results suggest that basal respiration in Foxp3$^+$IL-17A$^+$ cells is largely uncoupled from phosphorylation of ADP to ATP (Supplementary Fig. 5c). Therefore, unlike Foxp3$^+$ T$_{reg}$ cells, Foxp3$^+$ Th17$^+$ cells exert active aerobic glycolysis, demonstrating that the Foxp3-associated programme does not control the metabolic characteristics in Th cells.

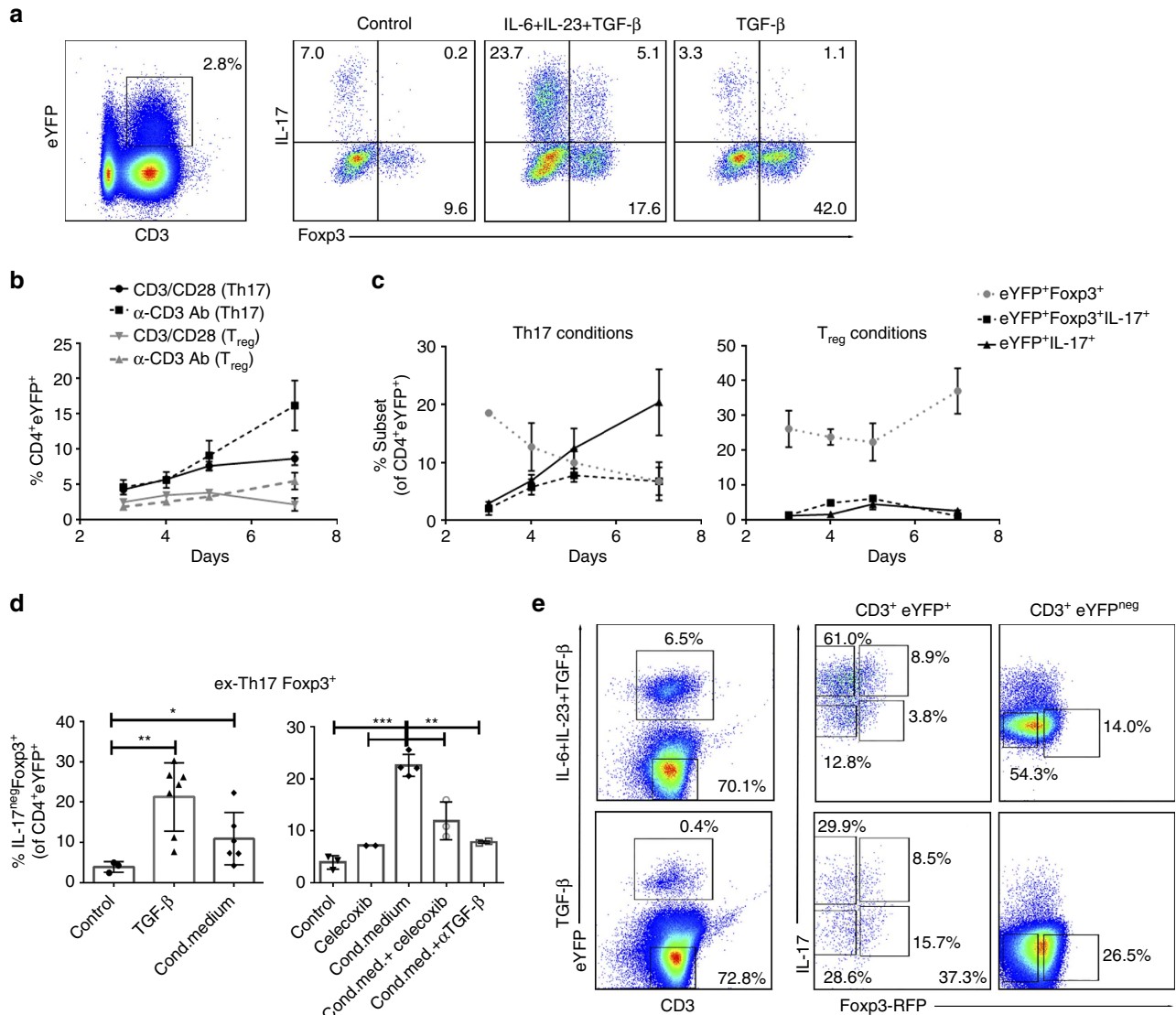

**Figure 4 | Tumour-associated TGF-β and PGE₂ promote ex-Th17 Foxp3⁺IL-17A^neg cells.** Splenocytes from *IL-17a^CreR26R^ReYFP* fate reporter mice were stimulated with CD3/CD28 microbeads under control, Th17 (IL-6, IL-23, TGF-β) and T_reg (TGF-β)-driving conditions. (**a–c**) Representative flow cytometry analysis (day 3, **a**) and time-dependent induction (**b,c**) of eYFP⁺ cells (**b**) and Th17–T_reg subsets (**c**) from IL-17a^CreR26R^eYFP fate reporter mice (CD4⁺ gated). Cells were cultured in the presence of Th17 (IL-6, IL-23 and TGF-β)- and T_reg (TGF-β)-driving conditions and analysed on days 3, 4, 5 and 7 by flow cytometry for the expression of eYFP. The fluorescence-activated cell sorting (FACS) gating strategy of live cells is presented in Supplementary Fig. 8e (**b**) and the expression of Foxp3 and IL-17A production by eYFP⁺ cells is shown. Similar results were obtained in an additional independent experiment. (**d**) CD4⁺ T cells from *IL-17a^CreR26R^ReYFP* reporter mice were cultured under T_reg (TGF-β)-driving conditions or conditioned medium of ID8A cells (aggregate data of 3–7 independent experiments; mean ± s.d., left). COX2 inhibitor celecoxib was added during preparation of conditioned medium and celecoxib and TGF-β blocking antibody were added to cell cultures during CD3/CD28 stimulation (right). (**e**) Representative flow cytometry analysis (day 7) of CD4⁺ T cells from *Th17^eYFP-Foxp3^mRFP* fate reporter mice cultured in the presence of Th17/T_reg (IL-6, IL-23 and TGF-β, 3 days and TGF-β, + 4 days)- and T_reg (TGF-β, 7 days)-driving conditions. The FACS gating strategy of live cells is presented in Supplementary Fig. 8e. All data are mean ± s.d. *P < 0.05, **P < 0.01 and ***P < 0.001.

Since T-cell function is thought to be dependent on metabolic reprogramming[49], we further evaluated whether the metabolic signature of Th17–T_reg subsets impacts their effector functions. Unexpectedly, despite having a different cell-intrinsic metabolic programme being highly glycolytic, YFP⁺Foxp3⁺IL-17A⁺ cells suppress CD4⁺ T-cell proliferation to a similar extent as YFP⁺Foxp3⁺IL-17A^neg T_reg cells (Fig. 5b and Supplementary Fig. 6). These data demonstrate that distinct glycolytic and oxidative phosphorylation programmes that are essential for effector and regulatory CD4⁺ T-cell subsets are independent of their suppressive functions. While Foxp3 imprints the immunosuppressive function, it does not control the metabolic phenotype of Th cells. Therefore, the inherent metabolism of Th

Foxp3⁺ T-cell subsets is not imposed by Foxp3 expression, nor is it indicative of their suppressive potential, but is likely associated to their inflammatory state.

The key role of metabolic cues and regulatory pathways in defining T-cell differentiation is underscored by observations that blocking glycolysis promotes T_reg cell generation[48]. We used 2-deoxy-D-glucose (2DG) and etomoxir, inhibitors of glycolysis and fatty acid oxidation, respectively, to study the effects of metabolic pathways in Th17-to-T_reg cell transdifferentiation. Our data reveal that blocking the glycolysis, being the prominent metabolic pathway in IL-17A-producing Foxp3^neg and Foxp3⁺ subsets, promotes Th17-to-T_reg cell transdifferentiation (Fig. 5c). While etomoxir predominantly inhibits IL-17A-production in

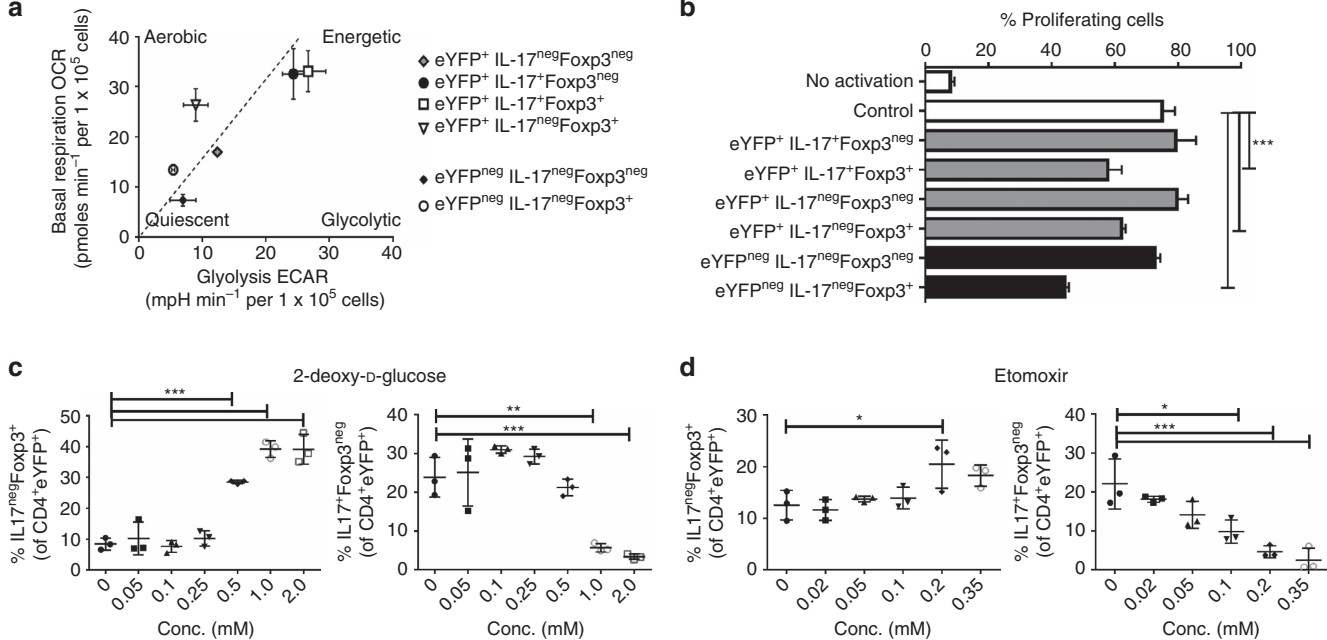

**Figure 5 | Suppressive Foxp3$^+$IL-17A$^+$ cells are metabolically active.** (**a,b**) YFP$^+$Foxp3$^{neg}$IL-17A$^+$, YFP$^+$Foxp3$^+$IL-17A$^+$, YFP$^+$Foxp3$^{neg}$IL-17A$^{neg}$, YFP$^+$Foxp3$^+$IL-17A$^{neg}$ and YFP$^{neg}$Foxp3$^{neg}$IL-17A$^{neg}$, YFP$^+$Foxp3$^{neg}$IL-17A$^{neg}$ CD3$^+$ T cells were sorted using the strategy presented in Fig. 4e and analysed with XFe Extracellular Flux Analyzer. (**a**) Cumulative data of two independent experiments evaluating basal respiration (OCR) versus glycolysis (basal extracellular acidification rate (ECAR)) of individual Th17–T$_{reg}$ subsets (mean values of each real-time run, $n = 2$ for YFP$^+$IL-17A$^+$Foxp3$^{neg}$, YFP$^+$IL-17A$^+$Foxp3$^+$, YFP$^+$IL-17A$^{neg}$Foxp3$^{neg}$, YFP$^+$IL-17A$^{neg}$Foxp3$^+$ and YFP$^{neg}$IL-17A$^{neg}$Foxp3$^+$, $n = 7$ YFP$^{neg}$IL-17A$^{neg}$Foxp3$^{neg}$). (**b**) CD4$^+$ T cells, isolated from Cd45.1 mice and stained with CFSE, were analysed after 72 h of stimulation with αCD3 Ab, in the presence of irradiated CD4$^{neg}$ fraction and Th17–T$_{reg}$ subsets, sorted as presented in Fig. 4e. One-way analysis of variance (ANOVA) of immunosuppressive effects of the subsets at 1:4 ratio subset:CD4$^+$. Similar results were obtained in an additional independent experiment. (**c,d**) The dose-dependent effects of 2DG (**c**) and etomoxir (**d**) on the transdifferentiation of Th17 cells (the percentage of IL-17A$^{neg}$Foxp3$^+$ cells of CD4$^+$eYFP$^+$) is shown. Similar results were obtained in an additional independent experiment. All data are mean ± s.d. *$P < 0.05$, **$P < 0.01$ and ***$P < 0.001$ by one-way ANOVA.

eYFP$^+$ cells and only promotes their conversion into Foxp3$^+$ T cells at high concentrations (350 μM etomoxir significantly affects the viability of cells, that is, 19.0% ± 5.6% of control) (Fig. 5d), the inhibition of glycolysis appears to primarily induce Foxp3 expression in IL-17A-producing cells, as evidenced by an increase in IL-17A$^+$Foxp3$^+$ cells of eYFP$^+$CD4$^+$ cells as well as an increase in exTh17 Foxp3$^+$ T cells with no concomitant decrease in IL-17A$^+$Foxp3$^{neg}$ cells with 0.5 mM 2DG (Fig. 5c).

The immunometabolism of the Th17–T$_{reg}$ subsets reveals an additional level of complexity in controlling the immune function of CD4$^+$ T cells. By metabolic reprogramming, it seems feasible to modulate (trans)differentiation of Th cells and thereby control their ultimate function and role in diverse environments.

**Th17-to-T$_{reg}$ cell transdifferentiation-associated targets.** The shared developmental pathways as well as the context-dependent plasticity and instability of Th17 and T$_{reg}$ cells provides an opportunity to identify mechanisms that specifically drive CD4$^+$ T-cell development towards Th17 and T$_{reg}$ cells, respectively. To this end, transcriptome analysis of the plastic subset may represent not only a strategy to identify unique features associated with the acquisition of immunosuppressive properties, but also to understand the molecular mechanisms behind Th17-to-T$_{reg}$ cell transdifferentiation. To identify unique gene expression profile associated with Th17 cell transdifferentiation, we analysed the DNA transcriptome of IL-17A$^+$Foxp3$^+$ cells (Fig. 6a and Supplementary Tables 1 and 2). Differential gene-level expression analysis with applied filters (analysis of variance $P < 0.05$ and linear fold change $< -2$ or $> 2$) reveals upregulation of 119 genes in the plastic Foxp3$^+$IL17A$^+$ cells compared with

Foxp3$^{neg}$IL-17A$^+$ Th17 cells (Fig. 6b)—that is, Th17–T$_{reg}$ plasticity markers. Of these, there are 15 coding genes of which the upregulated transmembrane molecules amenable for therapeutic interventions are folate receptor 4 (Folr4; 18.2-fold change), leucine-rich repeat containing 32 (Lrrc32, GARP; 16.96-fold change), peptidoglycan recognition protein 1 (Pglyrp1; 5.38-fold change), interleukin 1 receptor-like 1 (Il1rl1, ST2; 5.38-fold change), integrin, αE, epithelial-associated (Itgae; 4.7-fold change), T = cell immunoreceptor with Ig and ITIM domains (TIGIT; 4.21-fold change) and inducible T-cell co-stimulator (ICOS; 2.91-fold change) (Supplementary Table 1, Fig. 6b). We further validated the differential expression of Th17–T$_{reg}$ plasticity markers with flow cytometry (Fig. 6c and Supplementary Fig. 7). The recently reported specific reduction of T$_{reg}$ cells by Folr4 antibody, provoking effective tumour immunity in tumour-bearing animals[50], further substantiates the relevance of identified novel targets for antitumour interventions. Of additional therapeutic relevance may be transmembrane molecules-encoding complex genes integrin β8 (Itgb8), transmembrane protein 154 (TMEM154), CD86 antigen (Cd86), ectonucleoside triphosphate diphosphohydrolase 1 (Entpd1, CD39), intracellular targets (Helios, Aiolos) and non-coding genes for microRNA (Mir15b, Mir6933, Mir669d-2). In addition to the above-mentioned potential of the metabolic control of the Th17–T$_{reg}$ cell subsets, the differentially expressed targets identified by transcriptome analysis present the possibility of regulating the Th cell dynamics.

This novel targeting approach takes into account the plasticity of Th17 cells as an integral part of their biology. Th17 cells under different conditions may show distinct capacities for alternative fates depending on whether the (para)inflammatory conditions

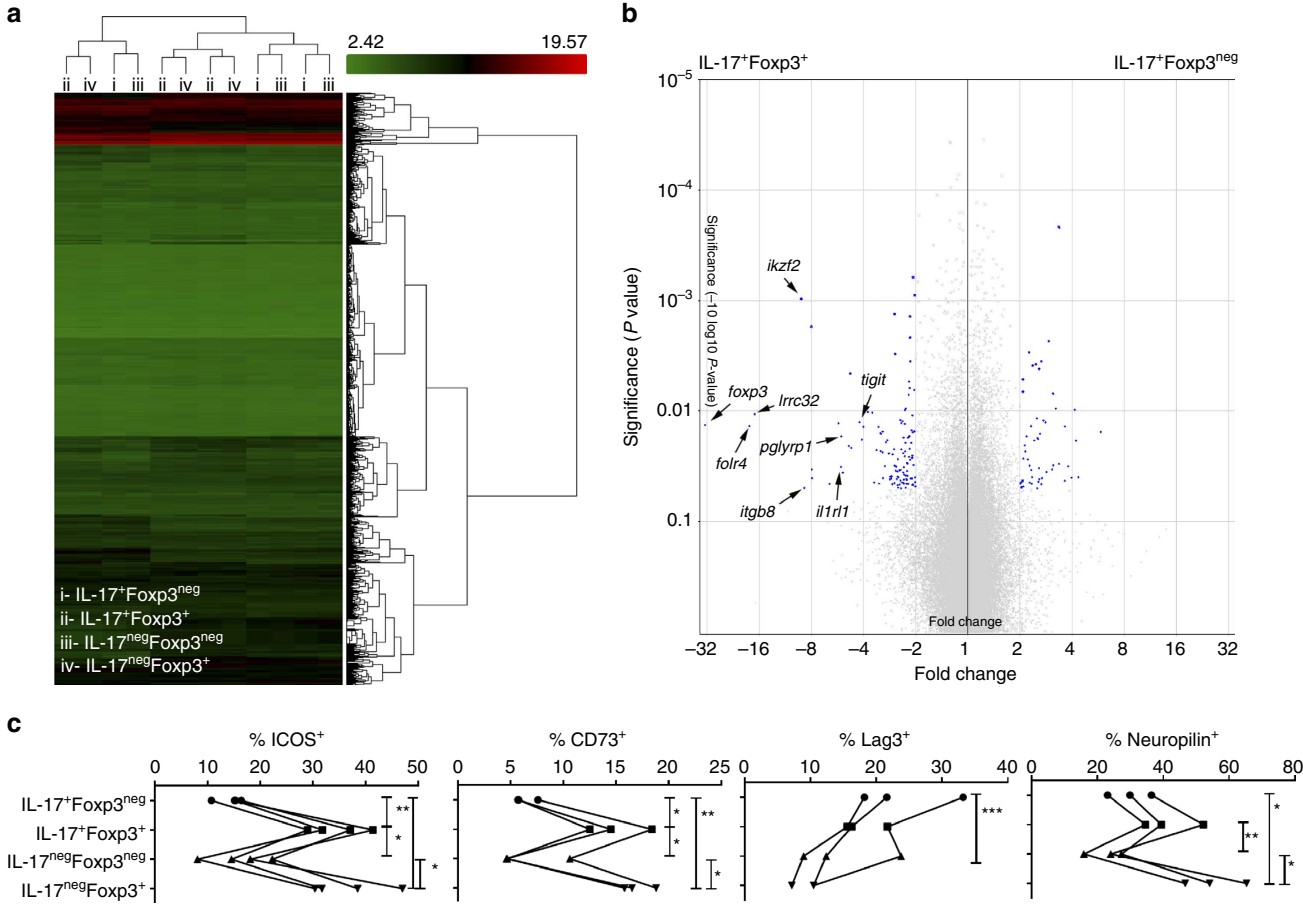

**Figure 6 | Transcriptome analysis of IL-17A$^+$Foxp3$^{+/neg}$ cells. (a–c)** Foxp3$^{neg}$IL-17A$^+$ (i), Foxp3$^+$IL-17A$^+$ (ii), Foxp3$^{neg}$IL-17A$^{neg}$ (iii) and Foxp3$^+$ IL-17A$^{neg}$ (iv) CD3$^+$ T cells were sorted using the strategy presented in Fig. 3a. **(a)** Heat map from hierarchical clustering of differentially expressed genes ($n=3$ per subset). **(b)** Volcano plot demonstrates increase in defined transmembrane markers (*Tigit, Tmem154, Folr4, Iikzf2, Pglyrp1, Cd86, Lrrc32, Il1rl1, Itgb8*) in Foxp3$^+$IL-17A$^+$ cells compared with Foxp3$^{neg}$IL-17A$^+$ T cells. **(c)** Cumulative data of flow cytometric staining (ICOS, CD73, Lag3, neuropilin1) of Th17-T$_{reg}$ subsets (i–iv) from three independent experiments. Additional data (TIGIT, ST2, Folr4, GARP, PGLYRP1) and representative flow cytometric staining are presented in Supplementary Fig. 7. All data are mean ± s.d. *$P<0.05$, **$P<0.01$ and ***$P<0.001$ by one-way analysis of variance (ANOVA).

are chronic and persistent or acute and rapidly resolve[13]. While induction of Foxp3 expression results in potent immunosuppression of IL-17A$^+$Foxp3$^+$ cells, the acquisition of functional characteristics of Th1 cells[20] (that is, production of interferon-γ) is necessary for potent antitumour activity of Th17 cells[51,52]. Therefore, we suggest that abrogating Th17-to-Foxp3$^+$ T-cell transdifferentiation in combination with prioritizing sequential (previously identified) Foxp3$^+$ to Th17 (ref. 53) to interferon-γ Th1-like[17,20] T-cell conversion is a viable method for T$_{reg}$ cell depletion. The inability of committed Th1 cells to acquire Th17–T$_{reg}$ features[30] results in an asymmetrical and constrained Foxp3$^+$-into-Th1/17 plasticity and could potentially result in final and irreversible Th1 cell polarization.

**Characteristics of Foxp3$^+$ IL-17A$^{+/-}$ cells in cancer patients.** A series of recent reports have documented a preferential expansion of tumour-promoting T$_{reg}$ cells exhibiting Th17 characteristics in the microenvironments of chronic inflammation and cancer[11,34,35]. Whether the origin of the plastic human tumour-associated IL-17A$^+$Foxp3$^+$ T cells are primarily IL-17A-producing or Foxp3-expressing progenitors remains unknown, ut most likely both pathways contribute to the development of pathogenic Foxp3-expressing Th17 cells, depending on diverse environmental inputs during cancer progression.

We analysed human ovarian cancer ascites for the presence of Th17–T$_{reg}$ subsets (Fig. 7). CD4$^+$ T cells infiltrating the ovarian cancer ascites demonstrate a propensity towards either Foxp3-expressing or IL-17A-producing phenotype (Fig. 7a), implying that the developmental pathways of T$_{reg}$ and Th17 cells are mutually exclusive. However, both of the phenotypes contain IL-17A$^+$Foxp3$^+$ T cells, which are significantly increased in the ovarian cancer ascites when compared with matched peripheral blood from the same ovarian cancer patients (Fig. 7b), indicating that Th17–T$_{reg}$ cell plasticity coincides with Th17–T$_{reg}$ cell imbalance in the human cancer microenvironment.

IL-17A$^+$Foxp3$^+$ T cells have been shown to phenotypically overlap with T$_{reg}$ and Th17 cells and express similar levels of CD25 and CCR4 (as T$_{reg}$ cells) and comparable levels of CD161 and CD49d (as Th17 cells)[34]. We further examined the expression of several Th17–T$_{reg}$ plasticity markers. Similar to the mouse IL-17A$^+$Foxp3$^+$ T cells, human ovarian cancer ascites-infiltrating IL-17A$^+$Foxp3$^+$ T cells show upregulation of neuropilin, GARP and ST2 compared with IL-17A$^+$Foxp3$^{neg}$ T cells, and present a phenotype similar to IL-17A$^{neg}$Foxp3$^+$ T cells (Fig. 7c). While some of the identified targets have been described on T$_{reg}$ cells, their relevance in tumour setting has not yet been evaluated. The inhibition of human T$_{reg}$ cell immunosuppressive activity by monoclonal antibodies against

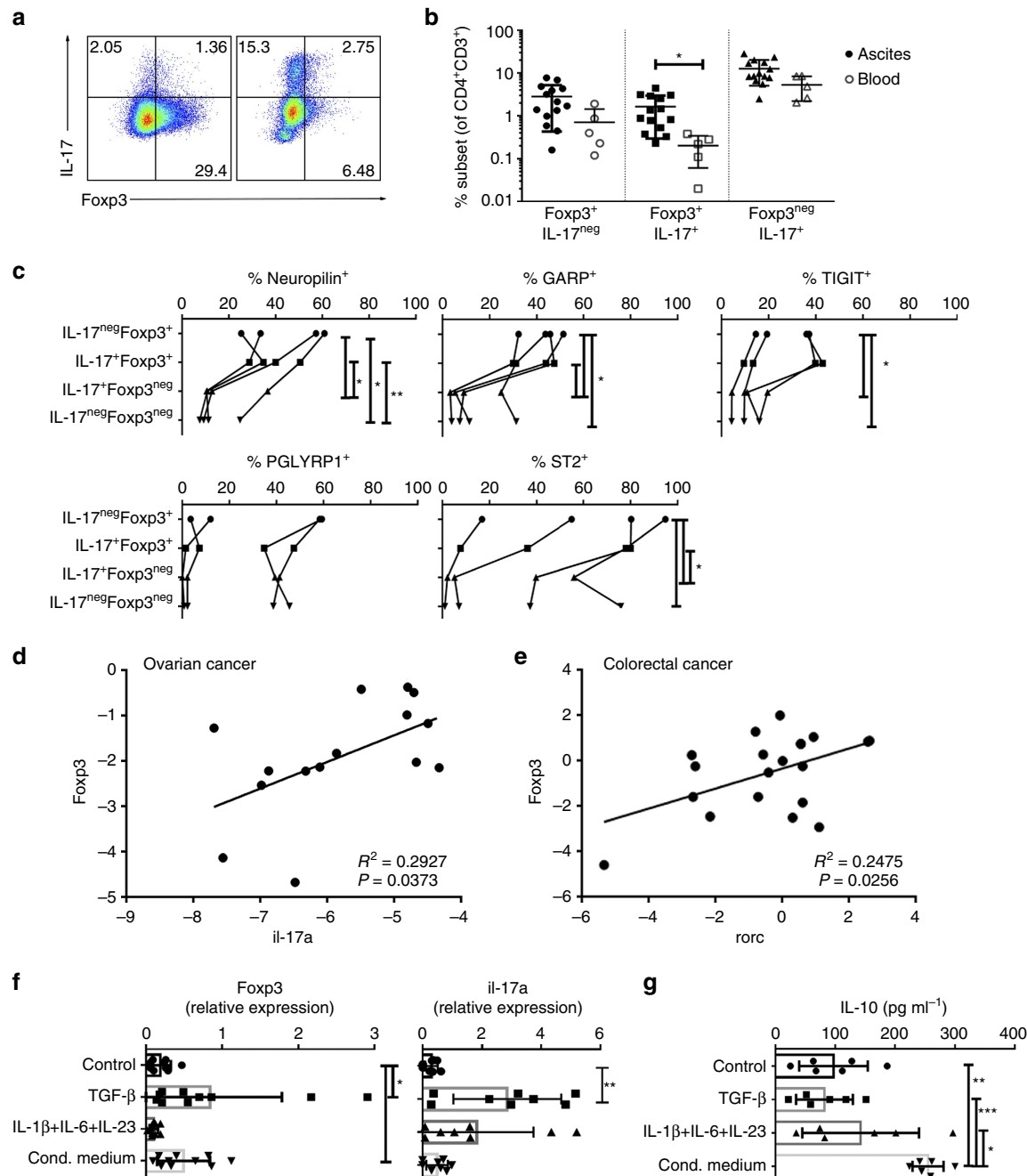

**Figure 7 | Tumour induction of $T_{reg}$-associated characteristics in human Th17 TALs.** (a–c) Foxp3-expressing IL-17A$^{+/-}$ cells from ovarian cancer patients are also characterized by expression of Th17–$T_{reg}$ cell plasticity-related targets. (a,b) Representative intracellular staining of IL-17A and Foxp3 in TALs (CD3$^+$CD4$^+$-gated) from two ovarian cancer ascites—note the inclination towards either Foxp3-expressing (left) or IL-17-producing (right) phenotype (a) and a comparison of the percentages of individual subsets (b) in TALs from ovarian cancer ascites ($n = 15$) compared with the lymphocytes from peripheral blood of the matched donors ($n = 5$). The fluorescence-activated cell sorting (FACS) gating strategy of live CD4$^+$ T cells is presented in Supplementary Fig. 8f. (c) Flow cytometric analysis of neuropilin, GARP, TIGIT, PGLYRP1 and ST2 expression in human ovarian cancer Th17–$T_{reg}$ CD4$^+$ TAL subsets ($n = 4$, refer to Table 1a for the details on patient data). All data are mean ± s.d. *$P < 0.05$ and **$P < 0.01$ by paired $t$-test. (d) Linear regression of $Il17a$ and $Foxp3$ expression in ovarian cancer TALs ($n = 15$, refer to Table 1b for the details on patient data). $R^2 = 0.2927$, $P = 0.0373$. (e) Linear regression of $Rorc$ and $Foxp3$ expression in colorectal cancer tissue ($n = 20$, refer to Table 1c for the details on patient data). $R^2 = 0.2475$, $P = 0.0256$. (f,g) Tumour-associated microenvironment induces $Foxp3$ expression and IL-10 production in human IL-17-producing CD4$^+$ cells. Expression of $Foxp3$ ($n = 10$) and $Il17a$ ($n = 8$) and IL-10 production ($n = 6$) by IL-17A-producing CD4$^+$ cells cultured in control, Th17- or $T_{reg}$-driving (TGF-β, ovarian cancer conditioned medium) conditions for 6 days. All data are mean ± s.d. *$P < 0.05$, **$P < 0.01$ and ***$P < 0.001$ by one-way analysis of variance (ANOVA).

GARP/TGFβ complexes[54] implicates these antibodies may serve as therapeutic tools to boost immune responses to cancer via a mechanism of action distinct from the currently available immunotherapies.

Our data further reveal a correlation of $Il17a$ and $Rorc$ with $Foxp3$ expression in ovarian cancer-associated and colorectal cancer-infiltrating cells (the expression of $Il17a$ in primary colorectal cancer cells is undetectable), respecitvely (Fig. 7d,e).

This correlation supports the concept that the developmental pathways of Th17 and $T_{reg}$ cells in cancer microenvironments are associated. This is also supported by the reduced percentage of tumour-associated Foxp3$^+$ T cells in ROR$\gamma$t mice and has previously been reported by Yang et al.[55]

**Induction of Foxp3$^+$ expression in human IL-17A$^+$ CD4$^+$ TALs.** There is a tight balance between Th17 and $T_{reg}$ cells. However, the susceptibility of human and mouse CD4$^+$ T cells to Th17-inducing factors is somewhat different[56–58]. While mouse Th17 cells are induced by a cytokine mixture including TGF-$\beta$ and IL-6, in humans, TGF-$\beta$ inhibits induction of Th17 cells which are induced by the cytokine mixture of IL-1$\beta$, IL-6 and IL-23 (ref. 59). In addition, i$T_{reg}$ cells are easily detected after CD4$^+$ T-cell stimulation with TGF-$\beta$ in mice, but not in humans. To address the relevance of human Th17 cell transdifferentiation into Foxp3-expressing cells, we cultured IL-17A$^+$CD4$^+$ TALs in $T_{reg}$-driving conditions. In line with the mouse data demonstrating the transdifferentiation of Th17 cells into IL-17$^{neg}$Foxp3$^+$ cells, Foxp3 expression is induced in human IL-17A-producing ovarian cancer TALs when CD4$^+$ IL-17$^+$ cells are restimulated under $T_{reg}$-driving conditions. Both TGF-$\beta$ and ovarian cancer conditioned medium upregulate Foxp3 expression in human CD4$^+$IL-17A$^+$ TALs compared with control and Th17-driving conditions (Fig. 7f). Interestingly, TGF-$\beta$, but not ovarian cancer conditioned medium, also promotes Il17a expression in CD4$^+$IL-17A$^+$ TALs, suggesting that TGF$\beta$ might be promoting IL-17A$^+$Foxp3$^+$ cells while the tumour microenvironment promotes exTh17 Foxp3$^+$ T cells. CD4$^+$IL-17A$^+$ TALs restimulated in ovarian cancer conditioned medium, but not in the presence of Th17-driving cytokines or TGF$\beta$, demonstrate increased production of IL-10 (Fig. 7g). Similarly, human Th17 clones have been shown to acquire IL-10-producing capacity while downregulating production of IL-17A in some cases when being continuously stimulated by antigen[60]. Altogether, these data illustrate the potential of human Th17 cells, like their mouse counterparts, to acquire $T_{reg}$-associated characteristics in a cancer microenvironment. Nevertheless, further analysis of the suppressive function and demethylation of the Foxp3 promoter is required to conclusively argue the conversion of human Th17 cells into $T_{reg}$ cells and address the (in)stability and plasticity of Th17-converted Foxp3-expressing cells. Recent data demonstrating that both strongly immunosuppressive Foxp3$^{hi}$ and weakly immunosuppressive Foxp3$^{lo}$ $T_{reg}$ TILs detected in colorectal cancer patients produce IL-17 further validate the relevance of such de novo Foxp3 expression in human IL17-producing T cells[61].

Th17 cells present a novel source of Foxp3$^+$ $T_{reg}$ cells in the setting of tumour. The identification of a Th17-to-exTh17 Foxp3$^+$ transdifferentiation pathway in the development of tumour-associated $T_{reg}$ cells warrants new approaches in targeting $T_{reg}$ cell-associated immunosuppression. Rather than eliminating detrimental $T_{reg}$ cells, targeting of the functional specialization of plastic and/or unstable Th17–$T_{reg}$ subsets allows for alternative manipulation of the $T_{reg}$-governed immune responses in cancer.

## Methods

**Mice.** C57BL/6 (B6), B6.SJL-Ptprc$^a$ Pepc$^b$/BoyJ (Cd45.1 mice), B6.129P2(Cg)-Rorc$^{tm2Litt}$/J (Ror$\gamma$$^{-/-}$), STOCK Il-17a$^{tm1.1(icre)Stck}$/J, B6.129 × 1-Gt(ROSA)26-Sor$^{tm1(EYFP)cos}$/J, B6.129(Cg)-Foxp3$^{tm3(DTR/GFP)Ayr}$/J (Foxp3$^{GFP}$) and C57BL/6-Foxp3$^{tm1Flv}$/J (Foxp3$^{mRFP}$) mice were obtained from The Jackson Laboratory. IL-17 fate reporter mice (Il17a$^{Cre}$Rosa26$^{eYFP}$) were created in our laboratory by breeding STOCK IL17a$^{tm1.1(icre)Stck}$/J and B6.129 × 1-Gt(ROSA)26Sor$^{tm1(EYFP)cos}$/J. Foxp3 fate reporter mice (Foxp3$^{YFP/Cre}$Rosa26$^{tdTomato}$) were created in our laboratory by breeding B6.129(Cg)-Foxp3$^{tm4(YFP/cre)Ayr}$/J and B6.Cg-Gt(ROSA)

26Sor$^{tm14(CAG-tdTomato)Hze}$/J. IL-17 fate reporter mice were bred with Foxp3$^{mRFP}$ mice to create Th17$^{eYFP}$-Foxp3$^{mRFP}$ fate reporter (Il17a$^{Cre}$Rosa26$^{eYFP}$ xFoxp3$^{tm1Flv}$) mice. All animal experiments were approved by the University of Pittsburgh Institutional Animal Care and Use Committee (15045926). Six- to eight-week-old female mice were used in the experiments. In all experiments tumour-bearing mice were randomly allocated to experimental groups. In experiments where in vivo immunological assays were performed, the variation within groups allowed the detection of differences with 4–5 mice per group.

**Mouse cell purification and tumour cell lines.** After removing spleens and lymph nodes of mice, single-cell suspensions (SCSs) were obtained by injecting phosphate buffer saline (PBS, Corning cellgro) into the spleens and mashing the organs over a 100 μm cell strainer (Fisher Scientific). Red blood cells were removed using ACK Lysis buffer (Life Technologies). The cells were washed with PBS, filtered through a 70 μm cell strainer (Falcon, Corning Incorporated) and then resuspended in PBS or culture medium. For the analysis of TALs, intraperitoneal fluid (i.p. washes or ascites) was collected. For the isolation of TILs, tumours were minced with two scalpels and incubated in a digestion buffer for 25 min at 37 °C and mashed over a pre-wetted 100 μm strainer. Red blood cells were removed with ACK Lysis buffer.

For mouse primary cell cultures, 100 U ml$^{-1}$ penicillin/streptomycin (Gibco, Invitrogen), 1 mM sodium pyruvate (Gibco, Invitrogen), non-essential amino acids (Sigma), 14.3 mM 2$\beta$-mercaptoethanol (Sigma-Aldrich), 2 mM L-glutamine (Gibco, Invitrogen) and 10% fetal bovine serum (FBS, Gemini Foundation B) were added to RPMI-1640 medium (Gibco, Invitrogen).

ID8A (luciferase-expressing), an aggressive cell line derived from spontaneous in vitro malignant transformation of C57BL/6 mouse ovarian surface epithelial cells, was a generous gift from Dr Tyler J Curiel. MC38 (C57BL/6J mouse strain) colorectal cancer cells (luciferase-expressing) were previously reported[62]. ID8A and MC38 colorectal cancer cells were cultured in Dulbecco's modified Eagle's medium (Gibco, Invitrogen) supplemented with 1% v/v (100 U ml$^{-1}$) penicillin/streptomycin, 2 mM L-glutamine and 10% FBS. The cell lines were regularly checked to ensure they are authentic and are not infected with mycoplasma.

**Analysis of in vivo Th17 cell transdifferentiation.** IL-17 fate reporter mice were injected i.p. with ID8A cells (4 × 10$^6$ cells per mouse) or MC38 cells (5 × 10$^5$ cells per mouse). For the ID8A experiment mice were killed on days 14, 21 and 29, while in the MC38 tumour model mice were killed on days 5, 12, 18 and 21. Typically untreated mice should be killed by day 40 ± 5 (ID8 model) or day 25 ± 5 (MC38 model). Spleens and ascites were collected and cells purified as described above. Cells were activated with phorbol 12-myristate 13-acetate (PMA, 50 ng ml$^{-1}$, Sigma) and ionomycin (1.0 μg ml$^{-1}$, Sigma) at 37 °C for 1 h before brefeldin (10 μg ml$^{-1}$, Sigma) was added for an additional 3 h. Cells were stained with Fixable Viability Dye-efluor 780 (eBioscience), anti-CD3-PE-Cy7 (6 μg ml$^{-1}$, eBioscience, clone: 145-2C11) and anti-CD4-PerCP-Cy5.5 (6 μg ml$^{-1}$, eBioscience, clone: RM4-5) and fixed with FoxP3 Fix/Perm Buffer Set (BioLegend) according to the manufacture's protocol. Intracellular staining was done using anti-IL-17A-PE (6 μg ml$^{-1}$, eBioscience, clone: eBio17B7) and anti-Foxp3-efluor 450 (6 μg ml$^{-1}$, eBioscience, clone: FJF-16s). Stainings were performed at 4 °C for 25 min. Cells were analysed by flow cytometry (LSRFortessa, BD Biosciences).

In addition, IL17A ELISA (R&D Systems) of cell culture supernatants (1 × 10$^5$ cells per well in a 96-well plate stimulated with Dynabeads Mouse T-Activator CD3/CD28 (Life Technologies)) was performed according to the manufacturer's protocol.

***Ex vivo* analysis of ROR$\gamma$t$^{-/-}$ mice.** B6 and Ror$\gamma$$^{-/-}$ mice were injected i.p. with ID8A-luc cells (4 × 10$^6$ cells per mouse). They were treated with either PBS (200 μl i.p.) or RMP1–14 (5 mg kg$^{-1}$, BioXcell, 200 μl i.p.) on days 3, 6, 9 and 12. Spleens and ascites from untreated B6 and Ror$\gamma$$^{-/-}$ mice (n = 5 per group) were collected at day 35 ± 2 and cells purified. Cells were stained with Fixable Viability Dye-efluor 780, anti-CD3-PE (6 μg ml$^{-1}$, eBioscience, clone: 145-2C11), anti-CD4-FITC (15 μg ml$^{-1}$, eBioscience, clone: RM4-4) and anti-PD1-PerCP-Cy5.5 (6 μg ml$^{-1}$, BioLegend, clone 29F.1A12). The cells were fixed with Foxp3 Fix/Perm Buffer Set and intracellular staining was performed using anti-Foxp3-APC (6 μg ml$^{-1}$, eBioscience, clone: FJK-16s) and anti-Helios-efluor 450 (3 μl per 100 μl, eBioscience, clone: 22F6). MDSCs were detected with anti-CD11b-efluor 450 (6 μg ml$^{-1}$, eBioscience, clone: M1/70) and anti-Gr-1-PE-Cy7 (6 μg ml$^{-1}$, eBioscience, clone: RB6-8C5).

In addition, Foxp3 mRNA (QT00138369) expression was analysed by TaqMan assay (LightCycler). The expression was normalized to the glyceraldehyde-3-phosphate dehydrogenase (Gadph) mRNA level and expressed as the relative expression, that is, fold increase (2$^{-DCT}$), where $\Delta CT = CT_{(Target gene)} - CT_{(GADPH)}$.

***In vitro* Th17–$T_{reg}$ cell differentiation.** CD4$^+$ T cells from IL-17 fate reporter mice were isolated using CD4 (L3T4) Microbeads (Miltenyi Biotec) according to the manufacturer's protocol. The negative fraction was also collected and irradiated (20 Gy) and mixed with CD4$^+$ cells (4:1). Cells were stimulated with either 1 μg ml$^{-1}$ anti-CD3e antibody (BD Bioscience, clone: 145-2C11) or Dynabeads Mouse T-Activator CD3/CD28 (3.5 μl ml$^{-1}$). Th17-driving cytokines (10 ng ml$^{-1}$

IL-23, 30 ng ml$^{-1}$ IL-6 and 2 ng ml$^{-1}$ TGF-β (all from R&D)), T$_{reg}$-driving cytokine (2 ng ml$^{-1}$ TGF-β) or conditioned medium of ID8A cells were added to cultures. To inhibit PGE$_2$ production, celecoxib (10 μM, Cayman Chemical) was added during preparation of the ID8A conditioned medium as well as to the T-cell cultures. Blocking of TGF-β was achieved with 0.2 μg ml$^{-1}$ anti-TGF-β blocking antibody (R&D Systems, clone: 9016). PGE$_2$ was used at 1 μM concentration. For inhibition of Th17–T$_{reg}$ differentiation, varying concentrations of etomoxir (20–350 μM) or 2DG (0.05–2.0 mM) were added to the cell cultures. Before staining, the cells were activated for 1 h with PMA (50 ng ml$^{-1}$, Sigma) and ionomycin (1 μg ml$^{-1}$, Sigma) at 37 °C. Brefeldin (10 μg ml$^{-1}$, Sigma) was added for additional 3 h of incubation. Cells were stained with Fixable Viability Dye-efluor 780, anti-CD3-PE-Cy7 (6 μg ml$^{-1}$, eBioscience clone: 145-2C11) and anti-CD4-PerCP-Cy5.5 (6 μg ml$^{-1}$, eBioscience, clone: RM4-5). The cells were fixed with Foxp3 Fix/Perm Buffer Set. Intracellular staining was done using anti-IL-17A-PE (6 μg ml$^{-1}$, eBioscience clone: eBio17B7) and anti-Foxp3-efluor 450 (6 μg ml$^{-1}$, eBioscience, clone: FJF-16s). Cells were analysed by flow cytometry (LSRFortessa, BD Biosciences).

**Generation of Th17–T$_{reg}$ subsets.** CD4$^+$ cells were isolated from SCSs of Foxp3$^{GFP}$ or Th17$^{eYFP}$-Foxp3$^{mRFP}$ reporter mice using CD4 (L3T4) Microbeads (SCSs from 3 to 6 mice were pooled together). The negative fraction was also collected, irradiated (20 Gy) and mixed with CD4$^+$ T cells (4:1 ratio). The cells were cultured under Th17-driving conditions (3.5 μl ml$^{-1}$ Dynabeads Mouse T-Activator CD3/CD28, 10 ng ml$^{-1}$ IL-23, 30 ng ml$^{-1}$ IL-6 and 2 ng ml$^{-1}$ TGF-β) for 72 h at 37 °C. Half of the medium was exchanged with T$_{reg}$-driving medium (2 ng ml$^{-1}$ TGF-β) for additional 24 h (Foxp3$^{GFP}$ subsets) or 4 day (Th17$^{eYFP}$-Foxp3$^{mRFP}$ subsets) incubation. The cells were then collected and CD3/CD28 Dynabeads removed. Cells were activated for 3 h with PMA (50 ng ml$^{-1}$) and ionomycin (0.75 μg ml$^{-1}$). A Mouse IL-17A Secretion Assay (Miltenyi Biotec Inc.) was used to detect IL-17A-secreting cells according to the manufacture's protocol. Cells from Foxp3$^{GFP}$ reporter mice were stained with anti-IL-17A/Biotin-PE (Miltenyi Biotec Inc., clone: Bio3-18E7) and anti-CD4-APC (4 μg ml$^{-1}$, eBioscience, clone: GK1.5) (Fig. 3b). Anti-IL-17A/Biotin-APC (Miltenyi Biotec Inc., clone: Bio3-18E7), in combination with anti-CD3-PE-Cy7 (4 μg ml$^{-1}$, eBioscience clone: 145-2C11), was used to stain the cells from Th17$^{eYFP}$-Foxp3$^{mRFP}$ reporter mice (Fig. 4e). DAPI (4,6-diamidino-2-phenylindole) was added before cell sorting (Beckman Coulter MoFlo Astrios). The Th17–T$_{reg}$ subsets were collected, resuspended in the culture medium and rested overnight at 37 °C.

**Adoptive transfer of Th17–T$_{reg}$ subsets.** Congenic Cd45.1 mice were injected i.p. with ID8A-luc cells (4 × 10$^6$ cells per mouse). Th17–T$_{reg}$ subsets (IL-17A$^+$ Foxp3$^{neg}$, IL-17A$^+$Foxp3$^+$, IL-17A$^{neg}$ Foxp3$^{neg}$ and IL-17A$^{neg}$Foxp3$^+$) were generated from Foxp3$^{GFP}$ reporter mice as described above and the same number of cells was injected i.p. (0.5–2.0 × 10$^5$ cells per mouse, n = 4 mice per group) on days 5, 12 and 19. Control mice were left untreated. Mice were imaged using the IVIS platform (Perkin Elmer). On days 40 ± 2, mice were killed to purify the cells from the spleens and the ascites. The cells were activated for 3 h with PMA (50 ng ml$^{-1}$) and ionomycin (0.75 μg ml$^{-1}$). A Mouse IL-17 Secretion Assay was used to detect IL-17-secreting cells. The cells were also stained for anti-CD4-APC (6 μg ml$^{-1}$, eBioscience, clone: GK1.5), anti-CD45.1-PE-Cy7 (6 μg ml$^{-1}$, eBioscience, clone: A20) and Fixable Viability Dye-efluor 780.

**Micro-immunosuppression assay.** IL-17A$^+$Foxp3$^{neg}$, IL-17A$^+$Foxp3$^+$, IL-17A$^{neg}$Foxp3$^{neg}$ and IL-17A$^{neg}$Foxp3$^+$ Th17-T$_{reg}$ subsets were generated from Foxp3$^{GFP}$ reporter mice and YFP$^+$IL-17A$^+$Foxp3$^{neg}$, YFP$^+$IL-17A$^+$Foxp3$^+$, YFP$^+$IL-17A$^{neg}$Foxp3$^{neg}$, YFP$^+$IL-17A$^{neg}$Foxp3$^+$, YFP$^{neg}$IL-17A$^{neg}$Foxp3$^{neg}$, YFP$^{neg}$IL-17A$^{neg}$Foxp3$^+$ Th17-T$_{reg}$ subsets were generated from Th17$^{eYFP}$-Foxp3$^{mRFP}$ reporter mice as described above. CD4$^+$ cells were isolated from Cd45.1 mice using CD4 (L3T4) Microbeads and stained with the proliferation dye, carboxyfluorescein succinimidyl ester (CFSE) according to the manufacturer's protocol (Molecular Probes). The negative fraction was also collected and then irradiated (20 Gy). The CD4$^+$ cells were activated with anti-CD3e antibody (1 μg ml$^{-1}$, BD Bioscience, clone: 145-2C11) in the presence of CD4$^{neg}$ fraction and Th17–T$_{reg}$ subsets (2:1–32:1 ratio CD4$^+$ target cells: Th17$^+$T$_{reg}$ subset). The cells were incubated for 72 h at 37 °C and then stained with Fixable Viability Dye-eFluor780, anti-CD45.1-eFluor450 (6 μg ml$^{-1}$, eBioscience, clone: A20) and anti-CD4-PerCP-Cy5.5 (6 μg ml$^{-1}$, eBioscience, clone: RM4-5). Cells were analysed by flow cytometry (LSRFortessa, BD Biosciences). The percentage of proliferating cells (CFSE$^{neg}$, gated on live/ CD45.1$^+$ CD4$^+$ cells) was calculated as a fraction of proliferating cells compared with control condition where CD4$^+$ cells were activated with anti-CD3e antibody (1 μg ml$^{-1}$, BD Bioscience, clone: 145-2C11) in the absence of Th17–T$_{reg}$ subsets.

**Metabolic assay.** IL-17A$^+$Foxp3$^{neg}$, IL-17A$^+$Foxp3$^+$, IL-17A$^{neg}$Foxp3$^{neg}$ and IL-17A$^{neg}$Foxp3$^+$ Th17–T$_{reg}$ subsets were generated from Foxp3$^{GFP}$ reporter mice and YFP$^+$IL-17A$^+$Foxp3$^{neg}$, YFP$^+$IL-17A$^+$Foxp3$^+$, YFP$^+$IL-17A$^{neg}$Foxp3$^{neg}$, YFP$^+$IL-17A$^{neg}$Foxp3$^+$, YFP$^{neg}$IL-17A$^{neg}$Foxp3$^{neg}$, YFP$^{neg}$IL-17A$^{neg}$Foxp3$^+$ Th17–T$_{reg}$ subsets were generated from Th17$^{eYFP}$-Foxp3$^{mRFP}$ reporter mice as

described above. After an overnight rest, the metabolic activity was determined using an XFe96 Extracellular Flux analyzer (Seahorse Bioscience). Briefly, 1 × 10$^5$ cells were seeded into CellTak (Corning) coated Seahorse microplates in unbuffered Dulbecco's modified Eagle's medium containing glutamine, glucose and pyruvate. After a brief rest for equilibration, readings were taken at 6 min intervals. Cells received four sequential injections of oligomycin (1 μM), FCCP (0.5 μM), 2DG (10 mM) and rotenone/antimycin A (100 μM) to obtain glycolytic and respiratory reserve and control values.

**Transcriptome analysis.** Th17–T$_{reg}$ subsets (IL-17A$^+$Foxp3$^{neg}$, IL-17A$^+$Foxp3$^+$, IL-17A$^{neg}$ Foxp3$^{neg}$ and IL-17A$^{neg}$ Foxp3$^+$) were generated from Foxp3$^{GFP}$ reporter mice as described above. Cell pellets were snap frozen on dry ice and kept at −80 °C until RNA extraction using Trizol Reagent following the manufacturer's protocol (Ambion/Life Technologies). Quality of RNA was assessed using Agilent RNA 6000 Pico Kit (Agilent 2100 Bioanalyzer). When possible, RNA concentration was determined using a Qubit 2.0 Fluorometer (Invitrogen/Life Technologies) and the Molecular Probes Qubit HS RNA Kit (Life Technologies) and for the other samples the concentration was estimated from the Pico Chip.

Gene expression was completed with Affymetrix WT Pico Kit (Affymetrix Mouse Transcriptome 1.0 Arrays) and GeneChip Hybridization, Wash and Stain Kit. The manufacturer protocols were followed for these processes. Quality control steps for the WT Pico amplification process included quantification on a Nanodrop 2000 (Thermo Scientific) and sizes were verified using Agilent 2100 Bioanalyzer (Agilent RNA 6000 Nano Kit). Quality was assessed for intake RNA, complementary RNA, unfragmented single-stranded complementary DNA as well as fragmented single-stranded complementary DNA.

**Immunophenotyping of Th17–T$_{reg}$ subsets.** Th17–T$_{reg}$ subsets (IL-17A$^+$ Foxp3$^{neg}$, IL-17A$^+$Foxp3$^+$, IL-17A$^{neg}$ Foxp3$^{neg}$ and IL-17A$^{neg}$ Foxp3$^+$) were stained with anti-neuropilin-1-PE-Cy7 (6 μg ml$^{-1}$, eBioscience, clone: 3DS304M), anti-Folr4-PE-Cy7 (6 μg ml$^{-1}$, eBioscience, clone: EBio12A5), anti-GARP-eFluor450 (6 μg ml$^{-1}$, eBioscience, clone: YGIC86), anti-ICOS-eFluor450 (6 μg ml$^{-1}$, eBioscience, clone: ISA-3), anti-Nrp1-PE-Cy7 (6 μg ml$^{-1}$, eBioscience, clone: 3DS304M), anti-ST2-PerCP-Cy5.5 (6 μg ml$^{-1}$, Biolegend, clone: DIH9), anti-TIGIT-PE-Cy7 (6 μg ml$^{-1}$, Biolegend, clone: 1G9), anti-CD73-PerCP-Cy5.5 (6 μg ml$^{-1}$, Biolegend, clone: TY/11.8) and anti-Lag3-eFluor710 (6 μg ml$^{-1}$, eBioscience, clone: eBioC9B7W). The cells were fixed with Foxp3 Fix/Perm Buffer Set and stained intracellularly with anti-Helios-eFluor450 (6 μg ml$^{-1}$, eBioscience, clone: 22F6). Cells were analysed by flow cytometry (LSRFortessa, BD Biosciences). The fluorescence minus one controls were used to identify the gating boundaries.

**Analysis of human cancer-associated Th17–T$_{reg}$ subsets.** Human ovarian cancer ascites were obtained intraoperatively from patients with primarily advanced (stage III or IV) epithelial ovarian cancer undergoing primary surgical debulking for clinical staging. All specimens (Table 1) were provided under protocols approved by the University of Pittsburgh or Roswell Park Cancer Institute institutional review boards (UPCI07-058 and CIC02-15) in accordance with the World Medical Association's Declaration of Helsinki, and written informed consent was obtained before any specimen collection. Analysis of mRNA expression was performed using the StepOne Plus System (Applied Biosystems), as previously described[63], using inventoried primer/probe sets. The expression of each gene was normalized to Hprt1 and expressed as fold increase (2$^{-ΔCT}$), where ΔCT = CT$_{(target\ gene)}$ − CT$_{(HPRT1)}$.

TaqMan analysis of mRNA expression in tumours and marginal tissues of colorectal cancer patients has been previously reported[64]. Tumour material (Table 1) was collected during routine surgery. All patients signed a consent approved by the institutional review board of the University of Pittsburgh for collection of tumour samples (UPCI 02-077).

For human primary cell cultures, RPMI-1640 medium (Gibco, Invitrogen) was supplemented with 8% human AB serum (Gemini Bio) instead of FBS. The ovarian cancer ascites TALs were isolated using density centrifugation with lymphocyte separation media. The cells were resuspended in culture medium (1 × 10$^6$ cells per ml) and activated with Dynabeads Human T-Activator CD3/CD28 (5.0 μl ml$^{-1}$, Life Technologies) for 6 days at 37 °C. The CD3/CD28 Dynabeads were removed and the cells activated with PMA (50 ng ml$^{-1}$) and ionomycin (1 μg ml$^{-1}$) for 3 h. Brefeldin (10 μg ml$^{-1}$) was added and the cells were incubated overnight. Following activation, cells were stained with Fixable Viability Dye-efluor 780, anti-CD4-PE-Cy7 (6 μg ml$^{-1}$, BD Bioscience, clone: SK3), anti-CD3-PerCP-Cy5.5 (6 μg ml$^{-1}$, BD Bioscience, clone: SP34-2), goat anti-hPGLYRP1 (3 μl per 100 μl, R&D systems, AF2590-SP), anti-neuropilin-1-APC (6 μg ml$^{-1}$, BioLegend, clone: 12C2), anti-GARP-APC (6 μg ml$^{-1}$, BioLegend, clone: 7B11), anti-hST2-APC (3 μl per 100 μl, R&D Systems, clone: 245707) and anti-hTIGIT-APC (3 μl per 100 μl, R&D Systems, clone: 741182). The cells stained with anti- hPGLYRP1 were subsequently stained with polyclonal anti-goat IgG-APC (3 μl per 100 μl, Abcam). The cells were fixed with Foxp3 Fix/Perm Buffer Set. Intracellular staining was performed using anti-IL-17A-PE (6 μg ml$^{-1}$, eBioscience, clone: eBio64CAP17), anti-Foxp3-pacific blue (6 μg ml$^{-1}$, BioLegend, clone: 206D) and anti-Helios-Alexa Fluor488 (6 μg ml$^{-1}$, BioLegend, clone:22F6). Cells were analysed by flow cytometry (LSRFortessa, BD Biosciences).

**Table 1 | Patient characteristics.**

**a. University of Pittsburgh Cancer Institute**

| ID | Age | Stage | Histology |
|---|---|---|---|
| TP14–893 | 70 | NA | Clear-cell adenocarcinoma of Mullerian origin |
| SB11–056 | 70 | NA | High-grade serous carcinoma |
| SB13–078 | 50 | IIIA | Clear-cell ovarian carcinoma |
| SB13–083 | 60 | IIIC | High-grade papillaty serous carcinoma of Mullerian origin |

**b. Roswell Park Cancer Institute**

| ID | Age | Stage | Grade | Histology | Debulk | Platinum | PFS | OS | Status |
|---|---|---|---|---|---|---|---|---|---|
| 20206 | 61 | IIIC | 3 | Serous | Suboptimal | Resistant | 14 | 22 | DOD |
| 17534 | 73 | IIIC | 3 | Mixed | Optimal | Refractory | 0 | 11 | DOD |
| 18275 | 66 | IIIC | 3 | Serous | Optimal | Sensitive | 18 | 71 | ANED |
| 19291 | 81 | IIIC | 3 | Undifferentiated | Optimal | Sensitive | 61 | 61 | ANED |
| RS112975 | 63 | IIIC | NA | Melanoma | Optimal | NA | 7 | 41 | DOD |
| 19933 | 59 | IIC | 2 | Serous | Optimal | Sensitive | 37 | 43 | AWD |
| RS1397573 | 78 | IIIC | 3 | Serous | Suboptimal | Refractory | 0 | 7 | DUC |
| 20289 | 82 | IIIC | 3 | Serous | Optimal | Refractory | 0 | 14 | DOD |
| 20625 | 68 | IV | 3 | Serous | Optimal | Sensitive | 10 | 27 | DOD |
| RS145218 | 66 | IIIC | 3 | Serous | Optimal | Sensitive | 39 | 39 | ANED |
| RS157897 | 44 | IIIC | 3 | Serous | Optimal | Sensitive | 16 | 37 | AWD |
| RS168195 | 38 | IIIC | 3 | Serous | Optimal | Resistant | 8 | 19 | DOD |
| RS976811 | 57 | IIIC | 3 | Serous | Optimal | Sensitive | 23 | 30 | AWD |
| RS967215 | 60 | IIIB | 3 | Serous | Optimal | Sensitive | 15 | 31 | AWD |
| RS1391920 | 68 | IIIC | 3 | Serous | Optimal | Refractory | 0 | 21 | DOD |

**c. University of Pittsburgh Cancer Institute**

| ID | Age | Sex | Stage | Histology |
|---|---|---|---|---|
| 47 | 65 | F | IV | Moderately differentiated adenocarcinoma |
| 52 | 69 | M | IV | Moderately differentiated adenocarcinoma |
| 96 | 69 | M | IV | Moderately differentiated, mucinous adenocarcinoma |
| 125 | 56 | M | IV | Well differentiated, mucinous adenocarcinoma |
| 142 | 60 | M | IV | Moderately differentiated adenocarcinoma |
| 145 | 50 | F | IV | Moderately differentiated adenocarcinoma |
| 146 | 38 | M | IV | Moderately differentiated adenocarcinoma |
| 153 | 67 | M | IV | Poorly differentiated, mucinous adenocarcinoma |
| 156 | 69 | M | IV | Moderately differentiated adenocarcinoma |
| 159 | 68 | M | IV | Moderately differentiated adenocarcinoma |
| 178 | 53 | F | IV | Well differentiated adenocarcinoma |
| 181 | 87 | F | IV | adenocarcinoma |
| 191 | 68 | M | IV | Moderately differentiated adenocarcinoma |
| 193 | 35 | M | IV | Mucinous adenocarcinoma |
| 195 | 68 | F | IV | Moderately differentiated adenocarcinoma |
| 199 | 68 | M | IV | Moderately differentiated adenocarcinoma |
| 200 | 68 | M | IV | Moderately differentiated adenocarcinoma |
| 201 | 59 | M | IV | Moderately differentiated adenocarcinoma |
| 209 | 60 | F | IV | Moderately differentiated adenocarcinoma |
| 211 | 44 | M | IV | Moderately differentiated adenocarcinoma |

Age, age at diagnosis; ANED, alive with no evidence of disease; AWD, alive with disease; Debulk, surgical debulking status is optimal (<1cm residual disease) or suboptimal (>1cm residual disease); DOD, died of disease; DUC, died of unknown cause; F, female; M, male; NA, not available; OS, overall survival (time from diagnosis to death or last follow-up); PFS, progression-free survival (time from diagnosis to first recurrence).
Platinum: resistant (recurrence <6 months from completing therapy), refractory (no disease-free interval) and sensitive (recurrence >6 months from completing therapy or no recurrence).
Human ovarian cancer ascites were obtained intraoperatively from previously untreated patients at the University of Pittsburgh Cancer Institute (a) and Roswell Park Cancer Institute (b) undergoing primary surgical debulking for clinical staging. Colorectal tumours and marginal tissues were collected during routine surgery at the University of Pittsburgh Cancer Institute (c). All patients signed a consent approved by the institutional review board of the University of Pittsburgh for collection of tumour samples (UPCI 02-077).

**In vitro restimulation of human IL-17A$^{+}$ CD4$^{+}$ TALs.** The ovarian cancer ascites cells were resuspended in culture medium ($5 \times 10^5$ cells per ml) and activated with Dynabeads Human T-Activator CD3/CD28 (3.5 µl ml$^{-1}$, Life Technologies) for 6 days at 37 °C. The CD3/CD28 Dynabeads were removed and the cells activated with PMA (50 ng ml$^{-1}$) and ionomycin (1 µg ml$^{-1}$) for 4 h. IL-17-producing cells were sorted (MoFlo Astrios, Beckman Coulter) following the labelling with IL-17 secretion assay-cell enrichment and detection kit (PE) (Miltenyi Biotec), anti-CD4-PECy7 (eBioscience) and DAPI. Sorted IL-17A-producing cells were resuspended in culture medium ($3–5 \times 10^5$ cells per ml) and activated with Dynabeads Human T-Activator CD3/CD28 (3.5 µl ml$^{-1}$, Life Technologies) in the absence or presence of 5 ng ml$^{-1}$ TGF-β, Th17-driving cytokines (20 ng ml$^{-1}$ IL-1β, 50 ng ml$^{-1}$ IL-6 and 10 ng ml$^{-1}$ IL-23) or ovarian cancer conditioned medium for 6 days at 37 °C. Ovarian cancer conditioned medium was prepared by culturing ovarian cancer ascites primary cells for 24 h in culture medium at $1 \times 10^6$ cells per ml. The medium was subsequently centrifuged for 10 min at 2,000 r.p.m. and supernatant, that is, ovarian cancer conditioned medium collected and stored until use. On day 6, supernatants were collected and analysed for IL-10 by ELISA (R&D Systems), and the cells lysed in RLT buffer for gene expression evaluation. Analysis of mRNA expression was performed using the StepOne Plus System (Applied Biosystems), as previously described[63], using inventoried primer/probe sets for human *Foxp3* and *Il17a*. The expression of each gene was normalized to *Hprt1* and expressed as fold increase ($2^{-\Delta CT}$), where $\Delta CT = CT_{(target\ gene)} - CT_{(HPRT1)}$.

**Data analysis.** All flow cytometry data were analysed with FlowJo (Treestar). Statistical analysis was performed using Prism 6.0 (Graphpad Software). Pearson's correlations between *Foxp3*, *Rorc* and *Il17a* were calculated on logarithmically

transformed data. Comparisons of continuous variables between groups were conducted using unpaired *t*-tests (two-tailed), one-way and two-way analysis of variance and linear regression according to the type of experiment. A paired *t*-test was used for the data shown in Fig. 7c. Survival curves in Fig. 2d were compared using log-rank (Mantel–Cox) test. The *P* values of $\leq 0.05$ were considered significant (*$P < 0.05$, **$P < 0.01$ and ***$P < 0.001$).

**Data availability.** The data that support the findings of this study are available from the corresponding author on request.

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

## Acknowledgements

This project used the University of Pittsburgh HSCRF Genomics Research service. The research was supported by NIH T32 CA113263 (to S.D.-C. and S.B.) and RPCI-UPCI Ovarian Cancer SPORE P50CA159981-01A1 (to N.O.). This study was supported in part by The David C. Koch Regional Therapy Cancer Center (to D.L.B.). We thank Dr Ravikumar Muthuswamy for providing the mRNA samples isolated from the tumour material of colorectal cancer patients and Kathryn Lemon for genotyping and monitoring the mice.

## Author contributions

S.D.-C., S.B., G.M.D. and N.O. performed the experiments; S.D.-C. and N.O. evaluated the experimental data; N.O. and D.L.B. designed the study; T.C. provided the reagents and the expertise and participated in manuscript preparation; G.M.D. and D.L.B. provided critical expertise and participated in manuscript preparation; K.O. and R.P.E. provided critical expertise and clinical material and participated in manuscript preparation; S.D.-C., S.B. and N.O. prepared the manuscript.

## Additional information

**Competing financial interests:** The authors declare no competing financial interests.

