## [Peer Review File · Nature Communications]

Reviewers' comments:

Reviewer #1, expert on Th17 and Tregs (Remarks to the Author):

In this manuscript Downs-Canner et al. aim to analyze the plasticity of Th17 cells in cancer. Overall the author show that Th17 cells in can give rise to Foxp3+ cells in tumor-bearing mice. They furthermore did transcriptome expression analysis and metabolic phenotyping of the different IL17A+/-Foxp3+/- T cell subsets. Overall this is an important study which represents a significant extension of recent publications showing Th17 plasticity.

The main caveat that I see is that the authors used a cancer model which is based on the transfer of tumor cells. It would be much better, if the authors could validate their results in a spontaneous or inducible cancer model. This would also allow the author to specifically analyze T cells isolated from the tumor or healthy tissue. Rather than performing the analysis in ascites and spleen.

An other caveat is the usage of in vitro differentiated Thelper cells in some of the key Figures. I would recommend the authors to validate at least some of their results shown in Figure 3,5,6 using ex vivo isolated T cells using the IL-17A Cre x Rosa YFP x Foxp3 mRFP reporter mice. This is a key part of the study, which is now mostly based on in vitro differentiated cells. But this does not really allow the dissection of Foxp3+ exTh17 cells from other Tregs.

In figure 4 the Foxp3 mRFP expression seems to be very low, compared to other published studies. Thus I would include a negative control (i.e. WT cells, which are Foxp3 mRFP negative) to validate the gating used.

Reviewer #2, expert on T cell lineage differentiation and plasticity (Remarks to the Author):

In the study, Downs-canner et al. demonstrates that Th17 cells are a novel source of tumor-induced Foxp3+ cells, indicating that tumor-driven Th17-into-Treg cell transdifferentiation could be novel targets in cancer immunotherapy. Th17 cells are prone to transdifferentiate into Treg cells, so this finding is not new. In 2005, Gagliani, N. et al. already showed that under different inflammatory disease, Th17-to-Treg cell transdifferentiation took place. Moreover, the manuscript has been written with a lot of type errors and it is not ready for publication in nature communication.

This study compared the cells between tumor ascites-infiltrating cells and spleen cells. It will be more informative to include tumor-infiltrating cells, since in the TEM, there is a lot of TGFb, which leads to Th17-into Treg cell transdifferentiation. In addition to ID8, they also took advantage of MC38 to establish murine CRC model. Although the data in CRC model is similar to those in ID8-bearing mice, they did not show the data about Th17-to-Treg transdifferentiation in CRC patients.

In the abstract part, the phenotype of ex-Th17 Foxp3+ cells (IL-17-FoxP3+ cells) should be defined.

In Fig. 1, the authors showed the kinetics of expression of Foxp3 and IL-17 in the eYFP+ cells. Other than Foxp3 and IL-17, they should also show the kinetics of RORyt to give a better idea about the transdifferentiation program.

The experiment shown in Fig.2, the sentence 'Foxp3+ cells infiltrating cancer ascites of RORgt-/- mice lacked the Helios+ subset (Fig. 2b)', which is misleading, since figure 2b showed that the expression of Helios in FoxP3+ cells from RORgt-/- mice was lower than that from wild type mice. In the knockout mice, Fig.2b only displayed reduced level of Helios. In addition, the author indicated that the lack of PD-1+ Treg cells was associated with a loss of therapeutic benefit from PD-1 blockade with treated WT mice outliving RORyt-/- mice. But do the expressions of PD-1 on

CD8 T cells and Foxp3⁻ CD4 T cells in the tumor from ROR γ t^{-/-} mice also change? If so, the loss of beneficial effects of PD-1 blockade is not solely due to decreased PD-1 on Treg.

For the immunosuppression assay, metabolic analysis and transcriptome analysis, the authors only focused on IL-17⁺Foxp3⁺ cells but neglected the exTh17 Foxp3⁺ cells. Although the author analyzed IL-17^{neg}Foxp3⁺ cells, but these cells are not necessarily exTh17, they can be nTreg or iTreg de novo differentiated from naïve T. It is of great importance to know whether these exTh17 Foxp3⁺ cells are also more suppressive and display some distinct metabolisms or molecular features since they outnumbered IL-17⁺Foxp3⁺ cells in the tumor and IL-17⁺Foxp3⁺ cells ultimately convert to exTh17 Foxp3⁺ cells.

The experiment shown in Fig.3C, it seemed that the adoptive transfer of FoxP+IL-17⁻ cells led to tumor regression, which should be explained. In general, Treg cells contribute to tumor progression.

The experiment shown in Fig.4d, what they found that TGF β was not the only factor produced by tumor cells promoting ex-Th17 Treg cells. PGE₂, another factor inducing Foxp3 expression and Treg cell function, proved to play a role in Th17-to-Treg cell conversion. They also used celecoxib and anti-TGF β separately to block each signaling pathway. How about the combination of those inhibitors?

The experiment shown in Fig. 6C should be better explained with regard to the gates used and the reason for the large difference in isotype control staining of ICOS and Helios.

Minor comments:

1. In Fig. 3d, the y axis title of the middle graph is missing.
2. In Fig. 5B, the y axis will be changed from glyolytic to glycolytic.
3. In Fig. 7a, the percentage of different populations should be added.
4. The second subtitle "Th17-associated Treg cell development pathway is supported by tumor-specific anomalies of Foxp3⁺ Treg cells in ROR γ t^{-/-} mice" does not make sense.

Reviewer #3, expert on ovarian cancer (Remarks to the Author):

May 19, 2016

In the paper by Downs-Canner et al., the authors investigate the ability of Th17 to transdifferentiate into different Treg subsets using primarily mouse models. In an elegant experiment they use an IL17 lineage tracing model to show that a population of formerly IL17⁺ lymphocytes transitions into an IL17⁻ Treg cell type. The authors show interesting metabolic and transcriptome differences between subsets. Overall, the data is interesting and contributes to the field but the clinical utility of abolishing the Foxp3⁺ IL17⁺ subset from tumors is not clear.

Major comments:

1. Fig 3c, the authors show that the Treg population iv seems to reduce tumor burden over the control. Is this significant? Please comment on this. Luminescence should be shown over time. Is there any change in tumor burden right after T cell infusion? Is PD-1 expression different between these subsets?
2. Fig 4d, The IL17⁺ population should be shown to prove that this is a specific effect on IL17⁻ Foxp3⁺ cells with these treatments. Also, are the left and right 2 distinct experiments? A control is needed for the right. In the text it's stated that PGE₂ is secreted by ID8 cells but this experiment does not prove that. PGE₂ could be released from the T cells upon stimulation with conditioned

media. Does recombinant PGE2 have this same effect?

3. Fig 4e, description and interpretation is missing from the main text.

4. Fig 7, Is there any correlation between Th17-Treg subsets and survival in the patients used. This would help to show clinical relevance. Are the T cells present within the tumors or only ascites? Tumor staining should be done.

Minor comments:

1. Many spelling errors need to be corrected

2. Expression of Foxp3 and IL17 should be consistently shown on the same axis (x or y) throughout all figures. Fig 4a is different than 3b.

3. Fig 2b and c, legend is needed for flow data

4. Methods and statistics need to be added for Fig 2d. How was the PD1 ab treatment done?

5. Fig 4b, legend is unclear, what is done in this experiment?

6. Fig 6a should be moved to supplement as it's not essential data

7. Fig 6b should be shown as a volcano plot to show significance and fold change

8. Fig 6c, data should be shown in bar graph format for each protein. It is very difficult to decipher the differences between subsets.

Reviewer #4, expert on T cells in cancer (Remarks to the Author):

The manuscript by Downs-Canner S. et al. describes tumor-driven Th17 differentiation into regulatory T cells (Tregs) in mouse models and proposes this as a possible target mechanism for cancer immunotherapy. Tregs play an important role in tumor immunity by suppressing a wide range of anti-tumor immune responses. High frequency of Tregs and their dominance to effector T cells in tumor tissues are associated with poor prognosis in various types of human cancers. Thus, developing strategy to control Tregs is an urgent issue in this field.

The authors addressed this issue and found that Th17 trans-differentiate into ex-Th17 Foxp3+Tregs upon tumor progression in two different mouse models. This increase of ex-Th17 Foxp3+Tregs was abrogated in Rorgt^{-/-} mice, indicating the important role of Rorgt for tumor-derived Treg induction. This Treg induction in tumors was dependent on TGF- β and PGE2. These Foxp3+ T cells including both IL-17-producing and IL-17-non-producing cells were immunosuppressive although they had totally different metabolic programs. Additionally, IL-17+Foxp3+Tregs harbored a specific gene signature which was associated with de novo Tregs. Furthermore, FoxP3-expressing IL-17+/- cells were also detected in ascites from ovarian cancer patients.

While addressing the trans-differentiation of Tregs from Th-17 is an interesting issue, several critical points were not addressed in this manuscript. Furthermore, there are major problems with the experimental design and interpretation of the data, particularly the differences in humans and mice (see below). This manuscript will probably not represent a major advance in the field.

Major point;

The authors employed mouse models for trans-differentiation from Th17 to Tregs. As a lot of studies have shown, the relationship between Th17 and Treg in humans and mice are totally different. For example, while Th17 are induced by a cytokine mixture including TGF- β and IL-6 in mice, this combination is not sufficient to induce Th17 in humans. In addition, induced Tregs are easily detected after CD4+ T-cell stimulation with TGF- β in mice, but not in humans. More importantly, it has been recently shown that human FoxP3+ cells composed of true Tregs and non-Tregs. These two different populations were detected in colorectal cancer patients and IL-17+ Tregs had suppressive function as observed in this manuscript, but their FoxP3 promoter region were totally demethylated, similar to de novo Tregs (Saito et al. Nat Med 2016 Apr 25). Therefore, my concern is that the phenomenon in this manuscript may be only observed in limited condition

of mouse models, but not in humans. Some experiments such as in vitro experiments needs to be confirmed with human materials. The presence of FoxP3-expressing IL-17+/- cells alone is not enough to show that this phenomenon is generally detected in human cancers.

Specific points;

1. In Figure 1 and 3 (particularly 3d), the authors show flowcytometry data. But the number of cells analyzed is too low to evaluate the data. For example, much higher numbers of spleen cells in Figure 1b can be prepared for the assay. Also the Foxp3-RFP gating in Figure 4e is difficult to be convinced. Actually almost every time the authors show representative data in every figure, but it is substandard and has no controls.

2. In Figure3, the number of cells transferred was very low and it should be difficult survive in animals without pre-conditioning. The authors need to explain why such small numbers of cells (shown in Figure 3d) were able to influence tumor growth.

3. In Figure 4, the importance of TGF-b and PGE2 in Treg induction was tested. A blocking assay with antibodies or gene targeting would strengthen the authors' conclusion.

4. In Figure 6b, gene expression was compared between IL-17+FoxP3+ and IL-17+Foxp3- cells. It should also be important to investigate specific gene expression pattern by comparing between IL-17+FoxP3+ and IL-17-Foxp3+ cells. Additionally, Figure 6a may be replaced with Supplemental Figure 5.

5. In Figure 2b and 2c, the explanation of flowcytometry data for filled-gray and black line is missing. The label of horizontal axis is missing in Figure 5d right.

Responses to Reviewers' comments:

Reviewer #1, expert on Th17 and T_{regs} (Remarks to the Author):

In this manuscript Downs-Canner et al. aim to analyze the plasticity of Th17 cells in cancer. Overall the author show that Th17 cells in can give rise to Foxp3⁺ cells in tumor-bearing mice. They furthermore did transcriptome expression analysis and metabolic phenotyping of the different IL17A^{+/-}Foxp3^{+/-} T cell subsets. Overall this is an important study which represents a significant extension of recent publications showing Th17 plasticity. The main caveat that I see is that the authors used a cancer model which is based on the transfer of tumor cells. It would be much better, if the authors could validate their results in a spontaneous or inducible cancer model. This would also allow the author to specifically analyze T cells isolated from the tumor or healthy tissue. Rather than performing the analysis in ascites and spleen.

We agree with the Reviewer that it would be beneficial to have the matching transdifferentiation of Th17 cells confirmed in a spontaneous or inducible mouse tumor model. We have explored the possibility to track the Th17 cell transdifferentiation in spontaneous triple transgenic mouse model (MUC1KrasPten) using conditional (Cre-loxP) mice with floxed genetic alterations in Kras (Kras^{G12D}) and Pten and expressing the human MUC1 antigen as self (Tirodkar et al. 2014 PlosONE; Zhang et al. 2016 Oncogene). By injection of Cre-encoding adenovirus (AdCre) in the ovarian bursa we activated the floxed mutations to initiate ovarian cancer, development of which is influenced by the anatomical environment. However, we were unable to overcome the technical hindrances to monitor the transdifferentiation of Th17 cells in this model. After the adoptive transfer of CD4⁺ splenocytes from *Il17a^{Cre}Rosa26^{eYFP}* mice intraperitoneally the number of T cells in the ovarian tumor at two different time points (4-weeks and 6-weeks) was too low to provide us any conclusive information. Apart from the spontaneous model, we have analyzed the healthy tissue (ovary and distal colon) from *Il17a^{Cre}Rosa26^{eYFP}* mice, however the infiltration of CD4⁺YFP⁺ T cells in these tissues is too low to allow monitoring the status of Th17 cell transdifferentiation.

To specifically analyze the T cells isolated from the tumor tissue that would substantiate the data from the cancer ascites tumor-associated lymphocytes, we performed novel experiment in *Il17a^{Cre}Rosa26^{eYFP}* mice (n=10) and have included the new data demonstrating the analysis of tumor-infiltrating lymphocytes (TILs) in murine CRC model (**Fig. 1**). Similar to our data in ovarian cancer-bearing mice, the data demonstrate the transdifferentiation of Th17 TILs into exTh17 Foxp3⁺ cells in colorectal cancer-bearing mice.

Another caveat is the usage of in vitro differentiated T helper cells in some of the key Figures. I would recommend the authors to validate at least some of their results shown in Figure 3,5,6 using ex vivo isolated T cells using the IL-17A Cre x Rosa YFP x Foxp3 mRFP reporter mice. This is a key part of the study, which is now mostly based on in vitro differentiated cells. But this does not really allow the dissection of Foxp3⁺ exTh17 cells from other T_{regs}.

We thank the Reviewer for this most relevant critique. We have performed additional experiments generating YFP⁺IL17A⁺Foxp3^{neg}, YFP⁺IL17A⁺Foxp3⁺, YFP⁺IL17A^{neg}Foxp3^{neg}, YFP⁺IL17A^{neg}Foxp3⁺, YFP^{neg}IL17A^{neg}Foxp3^{neg}, YFP^{neg}IL17A^{neg}Foxp3⁺ Th17-T_{reg} subsets from Th17^{eYFP}-Foxp3^{mRFP} reporter mice and have included the data demonstrating the metabolic characteristics (**Fig. 5a**), immunosuppressive function (**Fig. 5b**), and the expression of Helios and Ror γ t in the Foxp3⁺ exTh17 cells compared to other Th17-T_{reg} subsets (**Supplementary Fig. 4a**).

In figure 4 the Foxp3 mRFP expression seems to be very low, compared to other published studies. Thus I would include a negative control (i.e. WT cells, which are Foxp3 mRFP negative) to validate the gating used.

The expression of Foxp3 reporter proteins in Th17-T_{reg} subsets indeed seems to be low and this is something

we have addressed prior to initiating the experiments analyzing the individual Th17-T_{reg} subsets. The explanation for this is that flow cytometry detection of Foxp3 expression is technically challenging when the cells are concomitantly analyzed for IL17 production. Novel figure (see **Supplementary Fig. 8**) demonstrates that the reduced mRFP expression compared to published studies is the result of T cell activation with PMA and ionomycin for 3 h prior to staining, which is required for detection of IL17. When we analyze the cells which have not been activated, the mean fluorescence intensity (MFI) of mRFP is much higher. Also, we include the data comparing the expression of mRFP to GFP, YFP and dTomato (all Foxp3 reporters) to demonstrate that the MFI for all, mRFP, GFP and YFP are reduced following stimulation with PMA and ionomycin. In this experiment, CD4⁺ cells were activated with anti-CD3/CD28 microbeads in the presence of TGFβ and then analyzed by flow cytometry. Samples of each cell type was analyzed both without prior activation and after 4 hours of stimulation with PMA and ionomycin. As evident from the Table 1, *foxp3* is the highest induced gene in IL17⁺Foxp3⁺ cells compared to IL17⁺Foxp3^{neg} cells, which in our opinion sufficiently validates our gating strategy.

Reviewer #2, expert on T cell lineage differentiation and plasticity (Remarks to the Author):

In the study, Downs-Canner et al. demonstrate that Th17 cells are a novel source of tumor-induced Foxp3⁺ cells, indicating that tumor-driven Th17-into-T_{reg} cell transdifferentiation could be novel targets in cancer immunotherapy. Th17 cells are prone to transdifferentiate into T_{reg} cells, so this finding is not new. In 2005, Gagliani, N. et al. already showed that under different inflammatory disease, Th17-to-T_{reg} cell transdifferentiation took place. Moreover, the manuscript has been written with a lot of type errors and it is not ready for publication in Nature communication.

We thank the Reviewer for this comment. We agree with the Reviewer that Th17-into-T_{reg} cell transdifferentiation is not a novel phenomenon. We were the first to demonstrate direct Th17-into-T_{reg} cell transdifferentiation *in vivo* in a transplantation setting following mesenchymal stem cell therapy in combination with mycophenolate mofetil (Obermajer et al. *J Immunol.* 2014 – Ref.#13). Another exciting study recently reported Th17-into-T_{reg} cell transdifferentiation following resolution of inflammation (Gagliani et al. *Nature* 2015– Ref.#14). The novelty of the current study is that the tumor-associated immunosuppressive microenvironment induces Th17 cell transdifferentiation into exTh17 Foxp3⁺ T_{reg} cells. This is a novel pathway of tumor-associated T_{reg} cell emergence, with major implications regarding T_{reg} cell targeting in cancer immunotherapy.

This study compared the cells between tumor ascites-infiltrating cells and spleen cells. It will be more informative to include tumor-infiltrating cells, since in the TEM, there is a lot of TGFβ, which leads to Th17-into T_{reg} cell transdifferentiation. In addition to ID8, they also took advantage of MC38 to establish murine CRC model. Although the data in CRC model is similar to those in ID8-bearing mice, they did not show the data about Th17-to-T_{reg} transdifferentiation in CRC patients.

We have now included novel data demonstrating the analysis of tumor-infiltrating lymphocytes (TILs) in a murine CRC model. In analogy to our data in ovarian cancer-bearing mice, the novel data demonstrate the transdifferentiation of Th17 TILs into exTh17 Foxp3⁺ cells, which further substantiates Th17 cell transdifferentiation in a cancer setting.

Further, we now include additional data (**Fig. 7e**) demonstrating the correlation between *rorc* and *foxp3* in tumor samples from 20 CRC patients. Due to the restricted size of the tumors that we receive for research purposes, we were unable to analyze the IL17⁺Foxp3⁺ TILs in CRC patients by flow cytometry.

In the abstract part, the phenotype of ex-Th17 Foxp3⁺ cells (IL-17 FoxP3⁺ cells) should be defined.

We have now edited the abstract to include the information about the suppressive function and metabolic characteristic of ex-Th17 Foxp3⁺ cells.

In Fig. 1, the authors showed the kinetics of expression of Foxp3 and IL-17 in the eYFP⁺ cells. Other than Foxp3 and IL-17, they should also show the kinetics of RORγt to give a better idea about the transdifferentiation program.

We now include additional data demonstrating the kinetics of RORγt as well as Helios expression (**Supplementary Fig. 4b**) in YFP⁺ cells. In addition, we have included data showing the expression of RORγt and Helios in the plasticity subsets isolated from *Il17a^{Cre}Rosa26^{eYFP}Foxp3^{mRFP}* reporter mice (**Supplementary Fig. 4a**).

The experiment shown in Fig.2, the sentence 'Foxp3⁺ cells infiltrating cancer ascites of RORγt^{-/-} mice lacked the Helios⁺ subset (Fig. 2b)', which is misleading, since figure 2b showed that the expression of Helios in FoxP3⁺ cells from RORγt^{-/-} mice was lower than that from wild type mice. In the knockout mice, Fig.2b only displayed reduced level of Helios. In addition, the author indicated that the lack of PD-1⁺ T_{reg} cells was associated with a loss of therapeutic benefit from PD-1 blockade with treated WT mice outliving RORγt^{-/-} mice. But do the expressions of PD-1 on CD8 T cells and Foxp3⁻ CD4 T cells in the tumor from RORγt^{-/-} mice also change? If so, the loss of beneficial effects of PD-1 blockade is not solely due to decreased PD-1 on T_{reg}.

We have corrected the sentence to state that the percentage of Helios⁺ cells of the Foxp3⁺ CD4⁺ cells in the tumor ascites was reduced compared to the Foxp3⁺ CD4⁺ cells infiltrating the ascites of wild-type mice, as evident from the **Fig. 2b**.

The percentage of PD1⁺ cells of the CD4⁺Foxp3^{neg} cells was not significantly different between the TALs of wild-type and RORγt^{-/-} mice and was much lower than the CD4⁺Foxp3⁺ cells (mean 5.336 (control) and 5.472 (RORγt^{-/-}), n=5 for CD4⁺Foxp3^{neg} cells vs mean 42.4 (control) and 24.55 (RORγt^{-/-}), n=5 for CD4⁺Foxp3⁺ cells). Similarly, the percentage of PD1⁺ cells of the CD3⁺CD4^{neg} cells was not significantly different between the TALs of wild-type and RORγt^{-/-} mice (mean 21.48 (control) and 16.94 (RORγt^{-/-}), n=5). We have now included an additional narrative explaining that PD1 expression was significantly reduced specifically in CD4⁺Foxp3⁺ T_{reg} cells, but not CD4⁺Foxp3^{neg} cells or CD3⁺CD4^{neg} cells.

Also, we include additional data demonstrating the percentage of PD1⁺ cells of the Th17-T_{reg} subsets from *Il17a^{Cre}Rosa26^{eYFP}* reporter mice (**Supplementary Fig. 4c**).

For the immunosuppression assay, metabolic analysis and transcriptome analysis, the authors only focused on IL-17⁺ Foxp3⁺ cells but neglected the exTh17 Foxp3⁺ cells. Although the author analyzed IL-17^{neg} Foxp3⁺ cells, but these cells are not necessarily exTh17, they can be nTreg or iTreg de novo differentiated from naïve T. It is of great importance to know whether these exTh17 Foxp3⁺ cells are also more suppressive and display some distinct metabolisms or molecular features since they outnumbered IL-17⁺ Foxp3⁺ cells in the tumor and IL-17⁺ Foxp3⁺ cells ultimately convert to exTh17 Foxp3⁺ cells.

We agree with the Reviewer that the analysis of ex-Th17 Foxp3⁺ cells is critical for the interpretation of the role of exTh17 cells in the tumors. We have performed additional experiments generating YFP⁺IL17A⁺Foxp3^{neg}, YFP⁺IL17A⁺Foxp3⁺, YFP⁺IL17A^{neg}Foxp3^{neg}, YFP⁺IL17A^{neg}Foxp3⁺, YFP^{neg}IL17A^{neg}Foxp3^{neg}, YFP^{neg}IL17A^{neg}Foxp3⁺ Th17-T_{reg} subsets from Th17^{eYFP}-Foxp3^{mRFP} reporter mice and have included novel data in an extensively revised **Fig. 5**, demonstrating low levels of glycolysis in exTh17 Foxp3⁺ cells (**Fig. 5a**) and their immunosuppressive characteristics (**Fig. 5b**), which are similar to iT_{reg} cells.

The experiment shown in Fig.3C, it seemed that the adoptive transfer of FoxP3⁺IL-17 cells led to tumor regression, which should be explained. In general, T_{reg} cells contribute to tumor progression.

We agree with the Reviewer that the presented image of tumor progression on day 25 might give the impression that IL17^{neg}Foxp3⁺ cells are delaying tumor growth. However, the differences between the IL17^{neg}Foxp3⁺ treated mice and IL17^{neg}Foxp3^{neg} treated or control mice were not statistically significant. We decided to replace the **Fig. 3c** of the day 25 data with the day 32 data from the same experiment, where the data is similar and the message is the same, but the differences between untreated IL17^{neg}Foxp3^{neg} and IL17^{neg}Foxp3⁺ treated mice are not misleadingly apparent. Also, we include the tumor growth curves in **Supplementary Fig. 3a**.

The experiment shown in Fig.4d, what they found that TGFβ was not the only factor produced by tumor cells promoting ex-Th17 T_{reg} cells. PGE₂, another factor inducing Foxp3 expression and T_{reg} cell function, proved to play a role in Th17-to-T_{reg} cell conversion. They also used celecoxib and anti-TGFβ separately to block each signaling pathway. How about the combination of those inhibitors?

We have now performed additional experiments to study the effect of combinatorial inhibition of PGE₂ and TGFβ in T cell cultures stimulated in conditioned medium. Our data included in **Supplementary Fig. 4d** demonstrate that the combination of the inhibitors does not work additively or synergistically to inhibit Th17-to-T_{reg} cell conversion. Furthermore, while TGFβ by itself promotes Th17-to-T_{reg} cell conversion, TGFβ promotes IL17⁺ cells in the presence of IL6 and IL23, but the addition of PGE₂ reverses this Th17-driving potential of TGFβ to promote Th17-to-T_{reg} cell conversion. We have also included additional narrative to highlight the complexity of TGFβ and PGE₂ interplay in the transdifferentiation of Th17 cells.

The experiment shown in Fig. 6C should be better explained with regard to the gates used and the reason for the large difference in isotype control staining of ICOS and Helios.

We have now replaced the flow analysis data in **Fig. 6c** with the data in a bar graph format as well as added some additional data and included the staining for CD73 and TIGIT in an extensively revised **Supplementary Fig. 7b**. With this, we have changed the data for ICOS, so that the gate is similar to the one for Helios. The difference in the gates for the same fluorophores is due to different voltages when acquiring the samples. The gates were set based on the FMO staining (fluorescence minus one), where the samples were stained for all the markers but the one and this served to determine the negative population. We have now included additional information on how the gates were set in the *Methods* section, *Immunophenotyping of Th17-T_{reg} plasticity subsets*.

Minor comments:

1. In Fig. 3d, the y axis title of the middle graph is missing.

We have added the missing y axis title.

2. In Fig. 5b, the y axis will be changed from glyolytic to glycolytic.

We have corrected the spelling.

3. In Fig. 7a, the percentage of different populations should be added.

We have added the percentage of different populations in **Fig. 7a**.

4. The second subtitle "Th17-associated Treg cell development pathway is supported by tumor-specific anomalies of Foxp3+ Treg cells in RORγt^{-/-} mice" does not make sense.

We have changed the subtitle to "Reduced percentage and phenotypic changes of Foxp3⁺ T_{reg} cells in the tumor microenvironment of RORγt^{-/-} mice"

Reviewer #3, expert on ovarian cancer (Remarks to the Author): May 19, 2016

In the paper by Downs-Canner et al., the authors investigate the ability of Th17 to transdifferentiate into different T_{reg} subsets using primarily mouse models. In an elegant experiment they use an IL17 lineage tracing model to show that a population of formerly IL17⁺ lymphocytes transitions into an IL17⁺ T_{reg} cell type. The authors show interesting metabolic and transcriptome differences between subsets. Overall, the data is interesting and contributes to the field but the clinical utility of abolishing the Foxp3⁺ IL17⁺ subset from tumors is not clear.

Major comments:

1. Fig 3c, the authors show that the T_{reg} population iv seems to reduce tumor burden over the control. Is this significant? Please comment on this. Luminescence should be shown over time. Is there any change in tumor burden right after T cell infusion? Is PD-1 expression different between these subsets?

We have now included data demonstrating that the percentage of PD1⁺ cells of Foxp3^{neg}IL17⁺, Foxp3⁺IL17⁺ and IL17^{neg}Foxp3⁺ cells is not different among the three subsets, but is higher than on IL17^{neg}Foxp3^{neg} cells (**Supplementary Fig. 4c**).

We agree with the Reviewer that the presented image of tumor progression on day 25 might give the impression that IL17^{neg}Foxp3⁺ cells are delaying tumor growth. However, the differences between the IL17^{neg}Foxp3⁺ treated mice and IL17^{neg}Foxp3^{neg} treated or control mice were not significant. We decided to replace the **Fig. 3c** of the day 25 data with the day 32 data from the same experiment, where the data is similar and the message is the same, but the differences between untreated IL17^{neg}Foxp3^{neg} and IL17^{neg}Foxp3⁺ treated mice are not misleadingly apparent. Besides, we include the tumor progression curves in **Supplementary Fig. 3a**.

2. Fig 4d, The IL17⁺ population should be shown to prove that this is a specific effect on IL17 Foxp3⁺ cells with these treatments. Also, are the left and right 2 distinct experiments? A control is needed for the right. In the text it's stated that PGE₂ is secreted by ID8 cells but this experiment does not prove that. PGE₂ could be released from the T cells upon stimulation with conditioned media. Does recombinant PGE₂ have this same effect?

As suggested by the Reviewer we have included additional data to demonstrate the specificity of the effects of the inhibition of PGE₂ production and blockage of TGFβ. The data show that there is no significant effect of celecoxib and αTGFβ Ab on the IL17⁺Foxp3^{neg} cells (**Supplementary Fig. 4e**).

We included the control in **Fig. 4d**.

We thank the reviewer for the comment about the addition of celecoxib to block the PGE₂ production and have now edited the information in the *Methods* section so that it is clear that the celecoxib was present in the cultures of T cells in the condition medium.

Further, the additional data included in **Supplementary Fig. 4d** demonstrate that the addition of PGE₂ promotes Th17-to- T_{reg} cell conversion.

3. Fig 4e, description and interpretation is missing from the main text.

We have now added the description of the **Fig. 4e** in the main text.

4. Fig 7, Is there any correlation between Th17- T_{reg} subsets and survival in the patients used. This would help to show clinical relevance. Are the T cells present within the tumors or only ascites? Tumor staining should be done.

With the ascites samples analyzed we did not observe any correlation between the Th17- T_{reg} subsets and the survival of the patients, hence we did not include any narrative about the clinical relevance of the subsets.

We have now included additional data (**Fig. 7e**) demonstrating correlation between *rorc* (*il17a* was undetectable in CRC patient samples - we used 2 different primer sets to determine *il17a* expression) and *foxp3* expression in the tumor samples of 20 CRC patients.

Minor comments:

1. Many spelling errors need to be corrected.

We have corrected the spelling throughout the manuscript.

2. Expression of Foxp3 and IL17 should be consistently shown on the same axis (x or y) throughout all figures. Fig 4a is different than 3b.

We have now edited all the relevant figures to match the Foxp3 vs IL17 layout. (**Fig. 1b, Fig. 4a, Fig. 7a**).

3. Fig 2b and c, legend is needed for flow data.

We have included this information in the figure legend.

4. Methods and statistics need to be added for Fig 2d. How was the PD1 ab treatment done?

We have put additional information about the treatment into the figure legend. The statistics are now described in the **Methods, Data analysis**.

5. Fig 4b, legend is unclear, what is done in this experiment?

We have included additional narrative to explain what was done in the experiment presented in **Fig. 4b,c**.

6. Fig 6a should be moved to supplement as it is not essential data.

As per Reviewer #4, we have replaced the **Fig. 6a** with the **Supplementary Fig. 5**.

7. Fig 6b should be shown as a volcano plot to show significance and fold change.

We have replaced the **Fig.6b** with volcano plot.

8. Fig 6c, data should be shown in bar graph format for each protein. It is very difficult to decipher the differences between subsets.

We have now replaced the flow analysis data in **Fig. 6c** with the data in a bar graph format and have put the images of the flow cytometry staining in the **Supplementary Fig. 7**.

Reviewer #4, expert on T cells in cancer (Remarks to the Author):

The manuscript by Downs-Canner S. et al. describes tumor-driven Th17 differentiation into regulatory T cells (T_{regs}) in mouse models and proposes this as a possible target mechanism for cancer immunotherapy. T_{regs} play an important role in tumor immunity by suppressing a wide range of anti-tumor immune responses. High frequency of T_{regs} and their dominance to effector T cells in tumor tissues are associated with poor prognosis in various types of human cancers. Thus, developing strategy to control T_{regs} is an urgent issue in this field.

The authors addressed this issue and found that Th17 trans-differentiate into ex-Th17 Foxp3⁺ T_{regs} upon tumor progression in two different mouse models. This increase of ex-Th17 Foxp3⁺ T_{regs} was abrogated in $Ror\gamma^{-/-}$ mice, indicating the important role of $Ror\gamma$ for tumor-derived T_{reg} induction. This Treg induction in tumors was dependent on TGF- β and PGE₂. These Foxp3⁺ T cells including both IL-17-producing and IL-17-non-producing cells were immunosuppressive although they had totally different metabolic

programs. Additionally, IL-17⁺ Foxp3⁺ T_{regs} harbored a specific gene signature which was associated with de novo T_{regs}. Furthermore, FoxP3-expressing IL-17^{+/-} cells were also detected in ascites from ovarian cancer patients.

While addressing the trans-differentiation of T_{regs} from Th-17 is an interesting issue, several critical points were not addressed in this manuscript. Furthermore, there are major problems with the experimental design and interpretation of the data, particularly the differences in humans and mice (see below). This manuscript will probably not represent a major advance in the field.

Major point:

The authors employed mouse models for trans-differentiation from Th17 to T_{regs}. As a lot of studies have shown, the relationship between Th17 and T_{reg} in humans and mice are totally different. For example, while Th17 are induced by a cytokine mixture including TGF- β and IL-6 in mice, this combination is not sufficient to induce Th17 in humans. In addition, induced T_{regs} are easily detected after CD4⁺ T-cell stimulation with TGF- β in mice, but not in humans. More importantly, it has been recently shown that human FoxP3⁺ cells composed of true T_{regs} and non-T_{regs}. These two different populations were detected in colorectal cancer patients and IL-17⁺ T_{regs} had suppressive function as observed in this manuscript, but their FoxP3 promoter region were totally demethylated, similar to de novo T_{regs}. (Saito et al. Nat Med 2016 Apr 25). Therefore, my concern is that the phenomenon in this manuscript may be only observed in limited condition of mouse models, but not in humans. Some experiments such as in vitro experiments needs to be confirmed with human materials. The presence of FoxP3-expressing IL-17^{+/-} cells alone is not enough to show that this phenomenon is generally detected in human cancers.

We agree with the Reviewer that mouse models, in studying T cell biology, do not fully recapitulate the specifics of human Th17-T_{reg} cells. However, to monitor Th17 cell transdifferentiation, only fate reporter mouse models offer meaningful and conclusive information. To address the relevance of human Th17 cell transdifferentiation, we have now performed additional experiments, where we sorted IL17-producing cells and cultured the IL17⁺ cells *in vitro* in control, Th17-driving and T_{reg}-driving (including primary ovarian cancer cell conditioned medium) conditions (**Fig. 7f**). Our data demonstrate induction of *foxp3* expression in IL17-producing TALs as well as induction of IL10-production by these cells in cancer-associated environment. We include additional narrative to highlight the supportive nature of these data for the relevance of Th17 cells in human setting.

Specific points:

1. In Figure 1 and 3 (particularly 3d), the authors show flow cytometry data. But the number of cells analyzed is too low to evaluate the data. For example, much higher numbers of spleen cells in Figure 1b can be prepared for the assay. Also the Foxp3-RFP gating in Figure 4e is difficult to be convinced. Actually almost every time the authors show representative data in every figure, but it is substandard and has no controls.

We agree with the Reviewer that the higher numbers of YFP⁺ cells in the *in vivo* models would be beneficial. However, the number of TILs/TALs that we could isolate from each tumor-bearing mouse was limited. For this exact reason, we were unable to obtain any conclusive data analyzing the samples from spontaneous mouse model and healthy tissues from *IL17A^{Cre}Rosa^{YFP}* reporter mice.

Monitoring of Th17 cell transdifferentiation is particularly technically challenging because the cells need to be restimulated to assess IL17 production, but this apparently interferes with the expression of *foxp3*. We have now included an additional figure (see **Supplementary Fig. 8**) demonstrating that the reduced mRFP expression compared to published studies is the result of cell activation with PMA and ionomycin for 3 hours prior to staining for IL17, which is required for the detection of IL17 production. When we analyze the cells which have not been activated, the mean fluorescence intensity (MFI) of mRFP is higher. Also,

we include the data comparing the expression of mRFP to GFP, YFP and dTomato (all Foxp3 reporters) to demonstrate that the MFI for all, mRFP, GFP and YFP are reduced following stimulation with PMA and ionomycin.

2. In Figure 3, the number of cells transferred was very low and it should be difficult survive in animals without pre-conditioning. The authors need to explain why such small numbers of cells (shown in Figure 3d) were able to influence tumor growth.

Indeed the number of adoptively transferred Th17-T_{reg} cell subsets was low and for this reason we performed 3 consecutive injections (each one week apart) instead of one single injection of the subset. As mentioned in the narrative the survival of the mice was not affected, however the tumor growth was significantly different between IL17⁺Foxp3⁺ and IL17⁺Foxp3^{neg} subset compared to IL17^{neg}Foxp3^{neg} subset-injected mice.

Regarding the survival of the transferred cells, we performed the analysis of the TALs and stained the cells with a viability dye to exclude any dead cells from the analysis. Our data indicate that CD4⁺CD45.1^{neg} cells were viable (**Fig. 3d**) when analyzed for IL17-production and Foxp3 expression.

3. In Figure 4, the importance of TGF- β and PGE₂ in T_{reg} induction was tested. A blocking assay with antibodies or gene targeting would strengthen the authors' conclusion.

We thank the Reviewer for this suggestion. We did perform the blockage of TGF β using an α -TGF β blocking Ab. Further, apart from blocking COX2, we now include additional data with PGE₂ to demonstrate its relevance in Th17-to-Treg cell transdifferentiation.

4. In Figure 6b, gene expression was compared between IL-17⁺FoxP3⁺ and IL-17⁺Foxp3⁻ cells. It should also be important to investigate specific gene expression pattern by comparing between IL-17⁺FoxP3⁺ and IL-17⁺Foxp3⁺ cells. Additionally, Figure 6a may be replaced with Supplemental Figure 5.

We include a novel **Table 2** listing the differentially expressed genes in IL17⁺Foxp3⁺ and IL17^{neg}Foxp3⁺ cells.

We have replaced the **Fig. 6a** with the former **Supplementary Fig. 5**.

5. In Figure 2b and 2c, the explanation of flow cytometry data for filled-gray and black line is missing. The label of horizontal axis is missing in Figure 5d right.

We thank the Reviewer for this note. We have included the missing information into the figure legend.

REVIEWERS' COMMENTS:

Reviewer #1 (Remarks to the Author):

The authors have successfully addressed all my comments and concerns.

Reviewer #3 (Remarks to the Author):

The authors have carefully addressed all our comments. I'm satisfied by what they did. Their findings are a significant advance for the field.

Reviewer #4 (Remarks to the Author):

The authors responded my points, but I was not persuaded in several points. Particularly, they found induction of FOXP3 expression in IL-17-producing TALs. As IL-10 production is not enough to conclude these cells are Treg cells, analyzing suppressive function or demethylation of FOXP3 promoter region is necessary. Furthermore, although the authors claim the difficulty to evaluate a large numbers of cells (specific point 1), the data must be a key point of this study and need to show a conclusive data.

Responses to the Reviewer 4:

The authors responded my points, but I was not persuaded in several points. Particularly, they found induction of FOXP3 expression in IL-17-producing TALs. As IL-10 production is not enough to conclude these cells are T_{reg} cells, analyzing suppressive function or demethylation of FOXP3 promoter region is necessary.

We would like to argue that – as pointed by the Reviewer 4 - we do demonstrate the foxp3 induction in IL17-producing TALs from ovarian cancer patients. Since foxp3 is the key T_{reg}-associated transcription factor, we believe this data is supportive of the Th17 cell transdifferentiation demonstrated in tumor bearing mice. While we do agree with the Reviewer 4 that the analysis of demethylation of the foxp3 promoter region is an important aspect in Th17 cell transdifferentiation and would like to include additional narrative pointing out the relevance of foxp3 methylation status for the stability of exTh17 cells, we believe the epigenetic fingerprint of Th17 cell transdifferentiation is beyond the scope of the current manuscript. We are currently planning and performing novel experiments addressing this very issue, however it will take us several months to conduct all the necessary murine experiments and analysis of the data as well as to validate the findings in the samples from cancer patients.

Furthermore, although the authors claim the difficulty to evaluate a large numbers of cells (specific point 1), the data must be a key point of this study and need to show a conclusive data.

While we do acknowledge the fact that the numbers of the cells are limited in **Fig. 1b** and **Fig. 3d**, we would like to stress that the novelty and significance of the data shown in these figures is primarily of qualitative and not quantitative nature. The mere realization that the

- eYFP⁺ cells within the tumor microenvironments of the tumor bearing mice do not produce IL17 and do express Foxp3 (**Fig. 1b**) and
- CD45.1^{neg} CD4⁺ Foxp3^{neg} IL17-producing cells adoptively transferred in tumor bearing mice stop producing IL17 and start expressing Foxp3 (**Fig. 3d**)

Is evidence as presented that the exTh17 T_{reg} cells arise in the tumor microenvironment.

Being the expert on T cells in cancer, the Reviewer 4 will appreciate that - as with antigen-specific T cells in the tumors - the identification rather than quantification of exTh17 IL17^{neg}Foxp3-expressing cells is the important message of the presented data for the field of tumor immunology.

Unequivocally, we do appreciate this comment and would be willing to revise the narrative to the relevant figures to clearly define the limitations of the current study.